# Microfibril-associated glycoprotein 4 forms octamers that mediate interactions with elastogenic proteins and cells

Michael R. Wozny[1,4], Valentin Nelea[1,2,4], Iram Fatima S. Siddiqui[1], Shaynah Wanga[3], Vivian de Waard[3], Mike Strauss [1,5] ✉ & Dieter P. Reinhardt [1,2,5] ✉

Microfibril-associated glycoprotein 4 (MFAP4) is a 36-kDa extracellular matrix glycoprotein with critical roles in organ fibrosis, chronic obstructive pulmonary disease, and cardiovascular disorders, including aortic aneurysms. MFAP4 multimerises and interacts with elastogenic proteins, including fibrillin-1 and tropoelastin, and with cells via integrins. Structural details of MFAP4 and its potential interfaces for these interactions are unknown. Here, we present a cryo-electron microscopy structure of human MFAP4. In the presence of calcium, MFAP4 assembles as an octamer, where two sets of homodimers constitute the top and bottom halves of each octamer. Each homodimer is linked together by an intermolecular disulphide bond. A C34S missense mutation prevents disulphide-bond formation between monomers but does not prevent octamer assembly. The atomic model, built into the 3.55 Å cryo-EM map, suggests that salt-bridge interactions mediate homo-dimer assembly, while non-polar residues form the interface between octamer halves. In the absence of calcium, an MFAP4 octamer dissociates into two tetramers. Binding studies with fibrillin-1, tropoelastin, LTBP4, and small fibulins show that MFAP4 has multiple surfaces for protein-protein interactions, most of which depend upon MFAP4 octamer assembly. The C34S mutation does not affect these protein interactions or cell interactions. MFAP4 assemblies with fibrillin-1 abrogate MFAP4 interactions with cells.

Microfibril-associated glycoprotein 4 (MFAP4) is a 36 kDa extracellular matrix glycoprotein present in elastic fibre-rich tissues[1,2]. MFAP4 consists of an N-terminal signal peptide, an RGD recognition site for cell surface integrin receptors, and a C-terminal fibrinogen-related domain (FReD), which constitutes the bulk of the protein (Fig. 1A). MFAP4 partakes in elastin fibre formation (elastogenesis) and in extracellular matrix organisation by protein-protein and cell-matrix interactions. Diseases associated with MFAP4 include Marfan syndrome,

cardiovascular disorders, chronic obstructive pulmonary disease, liver fibrosis and cirrhosis, Smith-Magenis syndrome, asthma, and cancer (for review see[3]).

Elastogenesis represents a highly hierarchical universal process that starts with fibronectin network formation, which is strictly required for the formation of "bead-on-the-string" fibrillin-1 containing microfibrils[4–7]. These microfibrils provide a cell surface-associated scaffold for the deposition of tropoelastin as nanofibres, which

[1]Faculty of Medicine and Health Sciences, McGill University, Montreal, QC, Canada. [2]Faculty of Dental Medicine and Oral Health Sciences, McGill University, Montreal, QC, Canada. [3]Amsterdam UMC location University of Amsterdam, Medical Biochemistry, Amsterdam, The Netherlands; Amsterdam Cardiovascular Sciences, Amsterdam, The Netherlands. [4]These authors contributed equally: Michael R. Wozny, Valentin Nelea. [5]These authors jointly supervised this work: Mike Strauss, Dieter P. Reinhardt. ✉e-mail: mike.strauss@mcgill.ca; dieter.reinhardt@mcgill.ca

**Fig. 1 | Intermolecular disulphide bonds form between C34 of MFAP4 but are not necessary for multimerisation. A** Schematic of MFAP4 sequence depicting the N-terminal signal peptide, cell surface integrin-binding RGD motif, and the fibrinogen-related domain (FReD) of human MFAP4. Cysteine residues, N-glycosylation, and C-mannosylation sites are indicated. In MFAP4$_{C34S}$, cysteine 34 is altered to serine. Schematic was adopted from Kanaan et al.[3] **B** Coomassie-stained gels after SDS-PAGE of MFAP4 and MFAP4$_{C34S}$ under non-reducing (left lanes) and reducing (right lanes) conditions. The gel was repeated ten times with similar results. **C** Hydrodynamic radii of MFAP4 (blue) and MFAP4$_{C34S}$ (red) in TBS/Ca$^{2+}$, measured by DLS. Shown is one representative experiment measured in technical triplicates of a total of $n = 5$ independent experiments with similar results. Means and standard errors of the mean of hydrodynamic radii are indicated. **D** AFM and negative stain TEM of MFAP4 (top panels) and MFAP4$_{C34S}$ (bottom panels). The analyses were performed with two MFAP4 and one MFAP4$_{C34S}$ purifications in TBS/Ca$^{2+}$. Images were taken from several sample locations. Scale bars are 10 nm for all images. **E** NaCl-dependent hydrodynamic radii of MFAP4 (blue) and MFAP4$_{C34S}$ (red), determined by DLS. Data points are mean values of hydrodynamic radii derived from $n = 4$ independent experiments each measured in technical triplicates. Error bars represent standard errors of the mean. Source data are provided as a Source Data file.

possibly interact with cells via integrins and heparan sulfate[8,9]. Several accessory proteins play crucial roles in elastic fibre formation and homoeostasis, including MFAP4, fibulin-4, fibulin-5, latent transforming growth factor beta binding protein-4 (LTBP4), and lysyl oxidases, among others[10–16]. Mature elastic fibres are comprised of an inner core of crosslinked elastin, an outer mantle of microfibrils, and an interface connecting both[4,17].

A series of studies have substantiated the role of MFAP4 in elastogenesis. It is present at the elastic fibre-microfibril interface[1,18], and it typically associates with fibrillin-containing microfibrils in many elastic tissues, including aorta, lung, and skin[2,10,18–20], but not with elastin-free microfibrils in the ciliary zonules of the eye or the kidney mesangium[18]. Adult mice with a genetic deletion of *Mfap4* show impaired lung elastic properties and emphysema-like alterations[21], and carotid artery ligation delayed neointima formation[22]. Protein expression levels are elevated in several diseases, including chronic obstructive pulmonary disease[23], abdominal aortic aneurysms[24], or hepatic fibrosis[25]. MFAP4 is

upregulated in the extracellular matrix of the ascending aortae from individuals with Marfan syndrome, and high circulating plasma MFAP4 levels were associated with type B aortic dissections in the descending thoracic aorta[26]. Glycoproteomic analysis identified increased and more diverse N-glycosylation patterns of MFAP4 in patients with Marfan syndrome compared to controls[26].

MFAP4 forms a disulphide-bonded dimer, that further assembles into higher oligomers[27]. However, the exact composition and identity of those oligomers remains controversial. Based on biochemical studies, Schlosser et al. reported MFAP4 tetramers of homodimers (i.e., octamers)[27], whereas Pilecki et al. determined trimeric and hexameric structures of homodimers (i.e., hexamers and dodecamers, respectively)[10]. Irrespective of the multimeric structure, the prevailing paradigm is that disulphide-bonded monomers represent the core assembly unit. Multimerisation of other elastogenic proteins has significant functional consequences, as described for fibrillin-1 and −2[7], as well as for fibulin-4 and LTBP-4[11,28]. Thus, it is predicted that the

multimeric state of MFAP4 determines its specific functions. Whether N-linked glycosylation at positions 87 and 137 or mannosylation at position 235 plays a role in MFAP4 oligomerisation and function remains to be established[26,29]. With respect to elastogenesis, MFAP4 directly interacts with tropoelastin and the elastin-specific cross-link desmosine in a calcium-dependent fashion and with fibrillin-1 and −2 as well as with lysyl oxidase in a calcium-independent manner[10,30]. In addition, MFAP4 promotes coacervation of tropoelastin[10], and interacts with elastogenic cells via an RGD motif that interacts with integrin αvβ3 and possibly αvβ5, thereby promoting cell migration and proliferation[22,30]. Based on homology with related FReD family proteins as well as on the analysis by the AlphaFold software, it is predicted that four of the five cysteines within MFAP4 form intramolecular disulphide bonds (Fig. 1A)[3,31,32], which leaves the possibility that the fifth N-terminal C34 residue participates in an intermolecular disulphide bond between MFAP4 monomers. However, this prediction requires experimental validation, and the relationship between MFAP4 oligomerisation and its specific functions is virtually unknown.

Despite important advances in understanding the structural and functional relationships of elastogenic proteins in general and MFAP4 in particular, critical molecular mechanisms that drive elastic fibre assembly remain unclear without sub-nanometre structural information. Here, we make use of current single particle analysis with cryogenic transmission electron microscopy (cryo-EM) techniques to investigate the MFAP4 ultrastructure, determine its atomic model structure and assess its macromolecular assembly. We combine these structural insights with biophysical experiments to reveal structure-function relationships with respect to elastogenic mechanisms.

## Results

### MFAP4 multimers do not require C34-mediated disulphide bonds

To investigate the oligomeric state of MFAP4, and whether C34 is necessary for intermolecular disulphide-bond formation within a dimer, we produced highly purified recombinant human MFAP4 and MFAP4$_{C34S}$ using HEK293 cells. As described previously by others[10], MFAP4 migrated in SDS gels as a ~72 kDa band (dimer) under non-reducing conditions and as a ~36 kDa band (monomer) under reducing conditions (Fig. 1B). In contrast, MFAP4$_{C34S}$ migrated as a ~36 kDa band under both non-reducing and reducing conditions (Fig. 1B). This data provides evidence that C34 is required and sufficient for intermolecular disulphide-bond formation between monomers to stabilise dimers. In solution under physiological buffer conditions (TBS/Ca²⁺), dynamic light scattering (DLS) of both MFAP4 and MFAP4$_{C34S}$ demonstrated a similar hydrodynamic radius of 6.9 ± 0.1 nm and 6.8 ± 0.2 nm, respectively (Fig. 1C). Based on DLS particle size calibration with standard proteins, this is consistent with a molecular mass of ~280 kDa, suggesting that both MFAP4 and MFAP4$_{C34S}$ assemble as a complex of eight monomers in the presence of Ca²⁺ (Supplementary Fig. 1A). Similarly, MFAP4 in solution dissociated into dimers upon treatment with 1% SDS, causing a shift in hydrodynamic radius to 3.7 ± 0.1 nm (Supplementary Fig. 1B). When MFAP4 was treated with 50 mM reducing dithiothreitol (DTT) and 1% SDS, dimers were reduced to monomers with a hydrodynamic radius of 2.7 ± 0.1 nm (Supplementary Fig. 1B). Irrespective of DTT treatment, MFAP4$_{C34S}$ dissociated into monomers in the presence of 1% SDS with a hydrodynamic radius of 2.6 ± 0.1 nm (Supplementary Fig. 1B). Relatively low levels of DTT (~3 mM) were required for 50% reduction of MFAP4 and conversion of dimers to monomers under physiological buffer conditions, suggesting a surface-located position for the intermolecular C34 disulphide bond (Supplementary Fig. 1C). When analysed by atomic force microscopy (AFM) or negative-stain transmission electron microscopy (TEM), both MFAP4 and MFAP4$_{C34S}$ appeared as round particles ~15 nm in diameter by AFM and ~12 nm in diameter by TEM (Fig. 1D); again, indicative of higher-order oligomeric assemblies. DLS analysis of

MFAP4 under high salt conditions (2 M NaCl at a physiological pH 7.4) reduced the hydrodynamic radius of particles to 6.2 ± 0.2 nm (Fig. 1E). Interestingly, increasing NaCl concentrations affected the hydrodynamic radius of MFAP4$_{C34S}$ more strongly than MFAP4 (5.3 ± 0.1 nm at 2 M NaCl) (Fig. 1E), suggesting that C34 stabilises MFAP4 assemblies but is not necessary for the formation of higher-ordered assemblies.

### MFAP4 dimers assemble to form an octamer

Single particle analysis of MFAP4 cryo-EM data revealed similar structures to those observed by AFM and negative-stain TEM (Fig. 2A). Initial 2D class averages consisted of two predominant views of four lobes (Fig. 2B); one where the four lobes are equidistant to each other with a spherical cavity between these lobes and a second arrangement in which the cavity between the lobes is elongated along a single axis. These two views are consistent with top/bottom vs side views of an octamer, which is elongated along one axis when viewed from its side. Furthermore, an octameric arrangement of MFAP4 was suggested by 2D class averages, which appeared to be projections of eight MFAP4 molecules, each perched at the vertex of a cube. The presented 2D classes were collectively used for initial reference generation and subsequent 3D reconstruction using C1 point group symmetry, confirming that MFAP4 organizes as an octameric structure. Further refinement of the 3D cryo-EM map with D2 symmetry produced a 3.55 Å resolution (masked Fourier shell correlation (FSC)$_{half-map}$ = 0.143, Supplementary Figs. 2 and 3) structure of octameric MFAP4 (Fig. 2C; Supplementary Movie 1). The cuboid assembly was elongated along the axis drawn between the top and bottom faces of this complex.

An AlphaFold2-predicted model of MFAP4 was positioned within this cryo-EM map and further refined through iterative manual and automatic adjustment of atomic coordinates using Coot[33] and Phenix[34]. With our map, we could accurately resolve the positions of residues Q37-A255 (fixed-radius masked crossed correlation score = 0.84; deposited as PDB: 8UN7 and EMDB: EMD-42394). We observed cryo-EM map density compatible with intramolecular disulphide bonds between the modelled positions of C41-C70 and C199-C212. Although the N-termini preceding Q37 are not resolved within the map model and are predicted to be disordered regions by AlphaFold2, we observed that the Q37 residues of each of the four MFAP4 molecules at the top and bottom of an octamer are positioned to extend towards each other and outwards from the macromolecular assembly (Supplementary Movie 1). This arrangement potentially positions four integrin-binding RGD motifs at each of the poles of the MFAP4 octamer (Supplementary Movie 1, orange residues). Furthermore, we observed that intermolecular disulphide bonds between C34 residues of different MFAP4 monomers are possible for this model when constrained by the cryo-EM map, although we do not resolve these residues within the cryo-EM map. Thus, our cryo-EM map-based model is restricted to Q37-A255 (PDB ID: 8UN7). Here we present an integrative model of cryo-EM and biochemical data, which includes C34, L35, and Q36 to reflect the dimerisation of MFAP4 under non-reducing conditions in SDS gels mediated by C34 (Fig. 2D; Supplementary Movie 1). In agreement with the disruption of MFAP4 dimers by treatment with low dithiothreitol concentrations in a physiological buffer (Supplementary Fig. 1C), these disulphide bonds are exposed to the solvent and are readily available for a reducing agent. Thus, the top and bottom halves of the MFAP4 octamer, defined across the central XY plane, can be described as two tetramers, each composed of two dimers with N-termini linked by C34 disulphide bonds.

Within the tetrameric top and bottom halves of the MFAP4 octamer, monomers interact through intra-tetrameric interfaces (Fig. 2E, F; Supplementary Movie 2), including potential salt-bridge interactions between interchain residues R105-D159 and R94-E158, hydrogen bonds between interchain residues A106-N152, G104-E158, G104-V154, and R94-E158, as well as non-bonded interchain interactions between R105-N152, G104-D159 and F103-E158. These interactions occur between

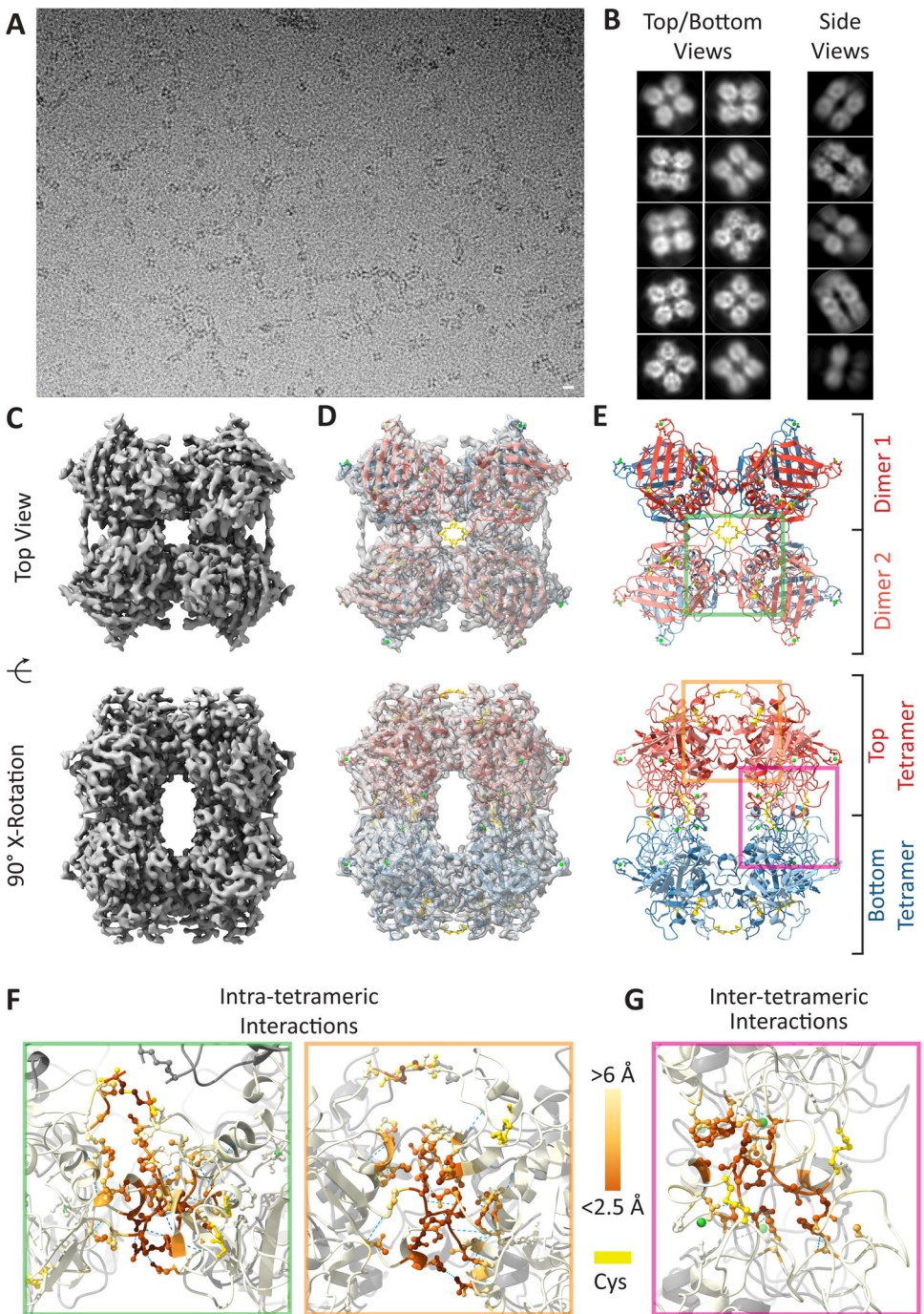

**Fig. 2 | Single particle analysis of MFAP4 with Ca²⁺.** **A** Cryo-EM micrograph of MFAP4 with $Ca^{2+}$ (scale bar 10 nm). Separate experiments were performed with two different MFAP4 purifications. **B** 2D-class averages of top/bottom and side views of MFAP4 particles used for reconstruction of initial 3D reference. 2.D classes were produced from particles extracted with a box size of 273.6 Å. **C** Top and side views of the D2 symmetry 3.55 Å resolution 3D cryo-EM map of MFAP4, **D** along with superimposed atomic model of Q37-A255 with integrative depiction of C34-Q36, and (**E**) the integrative atomic model (Q37-A255 resolved by cryo-EM map, inter-molecular disulphide bonds between C34 interpreted from presented biochemical and mutagenesis evidence, positions of C34-Q36 were modelled to minimise deviation from the expected bond length and angle restrictions in Coot 0.9.8.8). The top tetramer is red with the dimer halves coloured light and dark shades of red. The bottom tetramer is coloured blue with dimer halves coloured light and dark shades of blue. **F** Intra-tetrameric interactions as seen from the top (green inset) and side view (orange inset) of subpanel E. **G** Inter-tetrameric interactions as seen from the side view (pink inset) of subpanel E. Cysteine residues are coloured yellow in all subpanels. Hydrogen bonds are depicted as dashed blue lines. Residues are coloured according to inter-chain distance (dark orange <2.5 Å, light orange <8.5 Å; see colour scale in **F/G**). Source data are provided as a Source Data file.

each monomer so that there are four interaction surfaces comprised of salt-bridge, hydrogen bonds, and non-bonded interchain interactions per tetrameric half of an MFAP4 octamer. The buried solvent-accessible surface area – a measure of the extent of the interaction surface, or area not accessible to solvent – of these intra-tetrameric interaction surfaces is 615 or 633 Å² , depending on whether the monomers are disulphide bond linked. Each interface is shared between 40 contacting residues (16 residues of one chain interact with 24 residues of another chain). The top and bottom halves of the MFAP4 octamer associate through inter-tetrameric interfaces (Fig. 2G),

including a polar non-bonded interchain interaction between N231-A200 and non-polar non-bonded interchain interactions between V196-F93, L202-L202, S204-L202 (Fig. 2G; Supplementary Movie 3). Between tetrameric halves of the MFAP4 octamer, the buried solvent-accessible surface area is 433 Å$^2$ and is shared between 22 contacting residues (13 residues from each adjacent chain).

Noteworthy, these interfaces between tetramers correspond to regions within the previously described S1 binding site of MFAP4 (T187-K251)[10]. Previously resolved structures of other FReD proteins show that this site contains a Ca$^{2+}$-binding site. Ca$^{2+}$-autoradiography demonstrated that bovine MAGP-36 binds calcium[35]; however, one or more definitive Ca$^{2+}$-binding sites have not been resolved as alterations of potential Ca$^{2+}$-binding residues disrupted protein stability and expression[10]. We used ConSurf[36] to identify other conserved functional regions within the MFAP4 model (Supplementary Fig. 4B, C). Upon close inspection of loop A182-A206 as a candidate for Ca$^{2+}$-binding, we noticed cryo-EM map density between S186, T187, D191 and Q192 that was not explained by the atomic model of MFAP4. D191 and D193 of MFAP4 are conserved across several FreD proteins[10], and equivalent residues in FIBCD1 bind a Ca$^{2+}$ ion[31]. To account for this, we positioned a Ca$^{2+}$ ion within the cryo-EM map density near D191. We noticed that S186, D191, and Q192 sidechains, as well as the backbone carbonyl group of T187 were all capable of binding Ca$^{2+}$ at this position. We propose that this Ca$^{2+}$ helps to push loop A182-A206 away from MFAP4's centre of mass and towards the opposing A182-A206 loop of the other MFAP4 molecules, thus stabilising inter-tetrameric interactions between V196-F93, L202-L202, S204-L202, and N231-A200 to form octamers. We note that another conserved residue, D134, was observed to form a similar Ca$^{2+}$-binding pocket with E136 and N138. These residues are close to N137, which has previously been reported to be glycosylated[37], and thus may affect the stability of a glycan attached to N137.

## Structural evidence of glycosylation in MFAP4

Cryo-EM map densities reach out from the octamer at the positions N87 and N137, supporting that both sites are N-glycosylated (Supplementary Fig. 5A, B; Supplementary Movie 1, blue residues). We modelled a N-acetyl-D-glucosamine (NAG), the first unit of N-linked oligosaccharides, within the cryo-EM density nearby to both N87 and N137. Although the cryo-EM density at these positions extends beyond the modelled NAG molecules, as is common for variable glycosylation patterns, we were unable to fit further monosaccharides reliably in this extended domain. Only the first monosaccharide of the glycan at N137 can be discerned above the noise within the cryo-EM map, while the glycan at N87 is discernible for at least two of its constituent monosaccharides; although we only model one NAG at both N87 and N137. The cryo-EM density near N87 is visible at a higher contour than that around N137, suggesting that this glycan is less flexible or more common. It is positioned near the central cavity formed between tetramer halves of the MFAP4 octamer, while the N137 glycan is positioned at the periphery of the octamer. W235 was reported to be mannosylated and in the absence of this modification, MFAP4 secretion was decreased[29]. However, there is no cryo-EM map density around W235 that could correspond to a glycan (Supplementary Fig. 5C). W235 is buried within MFAP4 and not accessible as a surface residue. As such, W235 appears to play a structural role within the MFAP4 monomer and forms hydrogen bonds with A173, W238, and K239 through backbone-backbone interactions.

## MFAP4 octamers disassemble into tetramers without calcium

DLS measurements of MFAP4 and MFAP4$_{C34S}$ showed that hydrodynamic radii decreased from 6.9 ± 0.1 nm and 6.8 ± 0.2 nm in the Ca$^{2+}$-loaded form (Figs. 1C) to 5.3 ± 0.1 nm and 5.2 ± 0.2 nm after Ca$^{2+}$ was depleted with EDTA (Fig. 3A). Almost identical values were obtained after Ca$^{2+}$-depletion by EGTA (Fig. 3A). These values are consistent with

the possibility that ~280 kDa MFAP4 octamers in the Ca$^{2+}$ form dissociate into particles approximately half the molecular mass (~140 kDa) when Ca$^{2+}$ is removed (see calibration curve in Supplementary Fig. 1A). Interestingly, surface plasmon resonance spectroscopy (SPR) showed that MFAP4 octamers self-interact with a K$_D$ = 6.2 ± 0.9 nM in the presence of Ca$^{2+}$, but not after removal of Ca$^{2+}$ and possibly other divalent cations with EDTA (Fig. 3B). This data suggests that the conformation of the MFAP4 octamer is important for self-interaction. Similar self-interaction properties were observed for MFAP4$_{C34S}$ (K$_D$ = 5.8 ± 1.5 nM in the Ca$^{2+}$ form; no interaction in the Ca$^{2+}$-depleted form) (Fig. 3B), demonstrating that C34-mediated intermolecular disulphide bonds are neither required for self-interaction nor for the Ca$^{2+}$-mediated assembly of the MFAP4 octameric structure. The presence of Mg$^{2+}$ ions in the buffer could not rescue self-interaction of the proteins (Fig. 3B). Titrating cation-depleted MFAP4 or MFAP4$_{C34S}$ tetramers with Ca$^{2+}$ in DLS analyses showed dose-dependent formation of octamers, documented by increasing hydrodynamic radii from 5.0-5.3 to 6.8-7.1 nm (Fig. 3C). Contrary, Mg$^{2+}$ titration did not alter the tetrameric status (Fig. 3C). Similarly, adding other possibly relevant divalent cations (Zn$^{2+}$, Cu$^{2+}$, Mn$^{2+}$) at near physiological concentrations did not promote the formation of octamers from tetramers (Supplementary Fig. 6). Taken together this data demonstrate that Ca$^{2+}$ is exclusively required for the MFAP4 and MFAP4$_{C34S}$ assembly of tetramers into octamers and for self-interaction.

The shift in hydrodynamic radius of both MFAP4 and MFAP4$_{C34S}$ upon removal of Ca$^{2+}$ suggested a prominent structural change and prompted us to investigate Ca$^{2+}$-depleted MFAP4 by single particle analysis with cryo-EM. Micrographs of EDTA-treated MFAP4 showed structural assemblies similar to those of MFAP4 in the Ca$^{2+}$-form; however, the distribution of the particles was more evenly dispersed within the ice (Fig. 3D) as compared to MFAP4 with Ca$^{2+}$ (Fig. 2A). 2D classes of EDTA-treated MFAP4 resembled top/bottom view 2D classes of the MFAP4 octamer in the Ca$^{2+}$-form. However, potential side view 2D classes were absent from EDTA-treated MFAP4 particles (Fig. 3E). An asymmetric 3D reconstruction suggested a tetrameric structure. Refinement of the EDTA-treated MFAP4 cryo-EM map proved challenging, as it became clear that there was a strong preferred orientation of particles (~16 Å resolution, masked FSC$_{half-map}$ = 0.143, Supplementary Fig. 7). The refined tetramer structure of EDTA-treated MFAP4 fits within either top or bottom half of the Ca$^{2+}$-loaded MFAP4 octamer structure (Fig. 3F). Local resolution estimation using CryoRes[38,39] indicates that after B-factor sharpening with RELIONn4-beta these maps are ~9 Å resolution (Supplementary Fig. 8). We suspect that an increase in flexibility of the tetramer limits the resolution of the reconstruction. AFM analyses of EDTA-treated MFAP4 or MFAP4$_{C34S}$ showed particles consistent with top/bottom views, not with side views, of tetrameric particles (Fig. 3G). Importantly, however, the Ca$^{2+}$-depleted proteins demonstrated a significantly lower height (1.4 ± 0.2 nm for MFAP4; 1.5 ± 0.2 nm for MFAP4$_{C34S}$) compared to the Ca$^{2+}$-saturated proteins (3.5 ± 0.3 nm for both MFAP4 and MFAP4$_{C34S}$), supporting the conclusion that the EDTA-treated proteins represent half octamers, i.e., tetramers. Together, the data support a model where Ca$^{2+}$ is necessary for the assembly and/or stabilisation of octamers but not of tetramers. We propose that in the absence of inner Ca$^{2+}$-binding by S186, T187, D191, and Q192, the conformation of loop A182-A206 changes, thus preventing inter-tetrameric interactions between V196-F93, L202-L202, S204-L202 and N231-A200 to form octamers. Peripheral Ca$^{2+}$-binding by D134, E136, and N138 does not appear to be relevant to octamer formation.

## MFAP4 and MFAP4$_{C34S}$ interaction properties

Since MFAP4 associates with fibrillin-containing microfibrils and binds to fibrillin-1 and tropoelastin[10], we tested interactions with a series of elastogenic proteins and protein fragments using SPR to understand

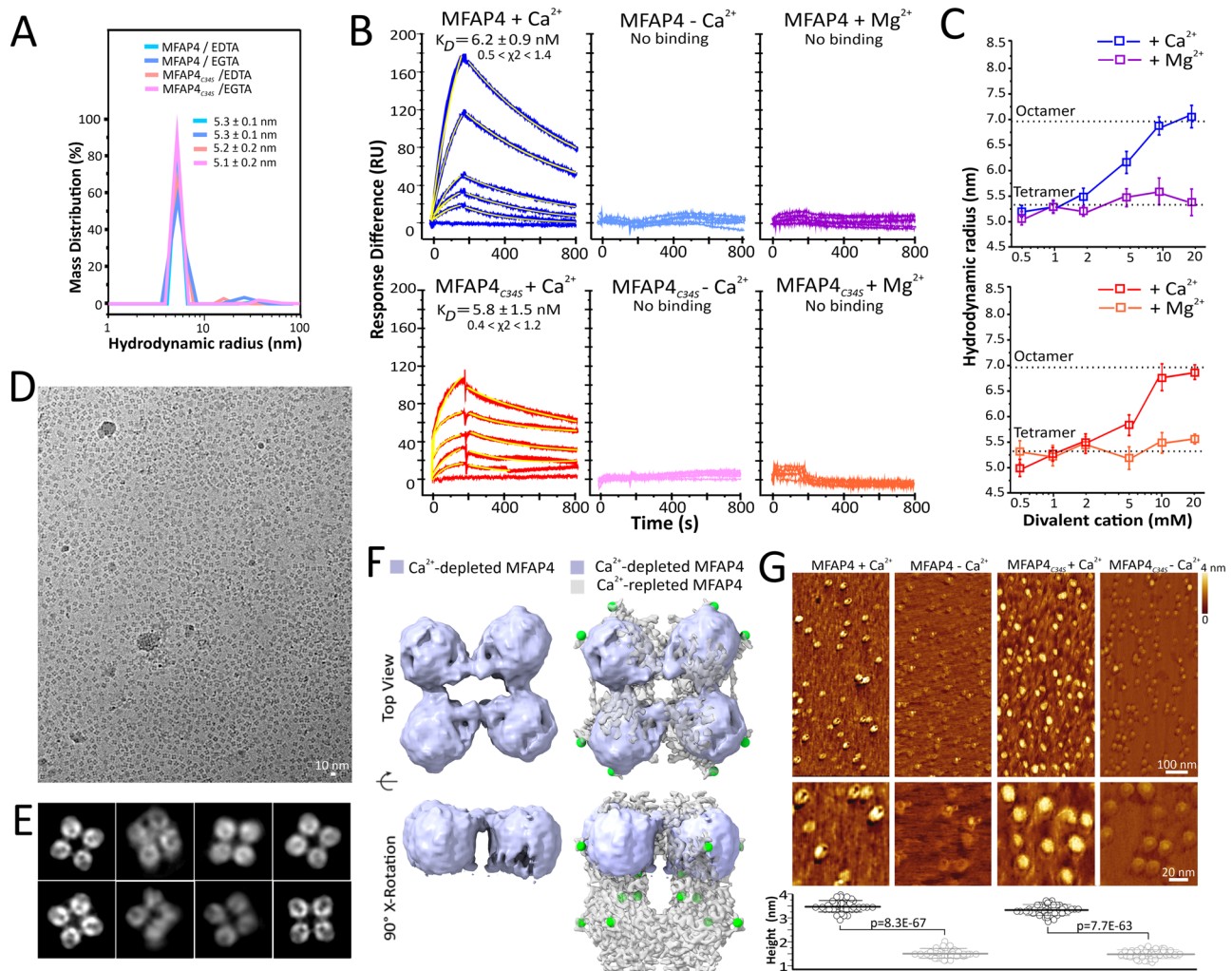

**Fig. 3 | Ca²⁺ is necessary for MFAP4 octamer formation and self-interactions.**
**A** Hydrodynamic radii of MFAP4 and MFAP4$_{C34S}$ in TBS without Ca²⁺ after treatment with EDTA or EGTA, measured by DLS. Shown is one representative experiment measured in technical triplicates of a total of $n = 5$ independent experiments with similar results. Means and standard errors of the mean of hydrodynamic radii are indicated. **B** Self-interaction of MFAP4 (top panels) and MFAP4$_{C34S}$ (bottom panels) measured by SPR with ( + Ca²⁺) or without (-Ca²⁺; EGTA treated) calcium, or with magnesium ( + Mg²⁺; EGTA treated). Each SPR profile corresponds to a different protein concentration (From top down: 100, 50, 20, 10, 5, and 0 μg/mL). The fittings are overlaid on the SPR traces in yellow. The χ2 value ranges (minimum-maximum) of the fittings are indicated. **C** DLS analyses of Ca²⁺-depleted (EGTA treatment) MFAP4 (top panel) and MFAP4$_{C34S}$ (bottom panel) with increasing concentrations of calcium ( + Ca²⁺) or magnesium ( + Mg²⁺). Data points are mean values of

hydrodynamic radii derived from $n = 4$ independent experiments each measured in technical triplicates. Error bars represent standard errors of the mean. (**D**) Cryo-EM micrograph of MFAP4 in TBS without Ca²⁺ after EDTA treatment (scale bar 10 nm). Experiments were performed separately with two MFAP4 purifications yielding similar results. **E** 2D-class averages of top/bottom of EDTA-treated MFAP4 (box size is 119.7 Å). **F** C2 3D reconstruction of EDTA-treated MFAP superimposed within the top half of the D4 reconstruction (grey) of MFAP4 with Ca²⁺(green). **G** AFM height images of MFAP4 and MFAP4$_{C34S}$ with ( + Ca²⁺) or without (-Ca²⁺; EDTA treated) calcium. Height quantification is shown on the bottom using $n = 51$ (MFAP4 + Ca²⁺), $n = 58$ (MFAP4$_{C34S}$ + Ca²⁺), $n = 55$ (MFAP4-Ca²⁺), and $n = 55$ (MFAP4$_{C34S}$-Ca²⁺) particles. Means and standard deviations are indicated. Statistical analysis was performed with the two-sample unpaired equal variance assumed t-test. p-values are indicated for each comparison. Source data are provided as a Source Data file.

how MFAP4 multimerisation and C34 are required for those interactions (Fig. 4 and Supplementary Table 1). In the presence of Ca²⁺ (octameric form), MFAP4 interacted with the N-terminal half (rFBN1-N, $K_D = 1.8 \pm 0.9$ nM) and with the centre (rF1M, $K_D = 1.2 \pm 0.5$ nM) of fibrillin-1 with very high affinity. Interactions with tropoelastin ($K_D = 45 \pm 9$ nM), LTBP4L ($K_D = 18 \pm 5$ nM), and LTBP4S ($K_D = 15 \pm 6$ nM) were of high affinity, the latter mediated by the N-terminal half of LTBP4L and LTBP4S (Supplementary Table 1). Binding to fibulin-4 ($K_D = 110 \pm 33$ nM) and fibulin-5 ($K_D = 357 \pm 46$ nM) were of lower affinity. Fibronectin, fibulin-3, and the C-terminal half of fibrillin-1 (rFBN1-C) did not interact with the Ca²⁺-loaded MFAP4 (Supplementary Table 1). After Ca²⁺ was removed selectively from MFAP4 (tetrameric form), it did not interact with the fibrillin-1 fragments, tropoelastin, fibulin-4, or fibulin-5, whereas MFAP4 interaction with LTBP4L and

LTBP4S was not affected by Ca²⁺-depletion of MFAP4 (Fig. 4). Control experiments in the presence of Mg²⁺ (Ca²⁺ removed with EGTA followed by addition of Mg²⁺) showed, as expected, that MFAP4 and MFAP4$_{C34S}$ did not interact with rFBN1-N or tropoelastin (Supplementary Fig. 9). Together, these data are consistent with the interpretation that the fully assembled MFAP4 octamer, requiring the presence of Ca²⁺, is essential for interaction with fibrillin-1, tropoelastin, fibulin-4, and fibulin-5, whereas the MFAP4 tetramer is sufficient to interact with LTBP4L and LTBP4S. For MFAP4$_{C34S}$, relatively similar binding properties were observed in the Ca²⁺ form, except for fibulin-5, which did not interact (Fig. 4). None of the ligands could interact with MFAP4$_{C34S}$ upon removal of Ca²⁺, including those interactions that occurred with MFAP4 in the Ca²⁺-depleted form (LTBP4L, LTBP4S) (Fig. 4). These data suggest that the C34-mediated intermolecular

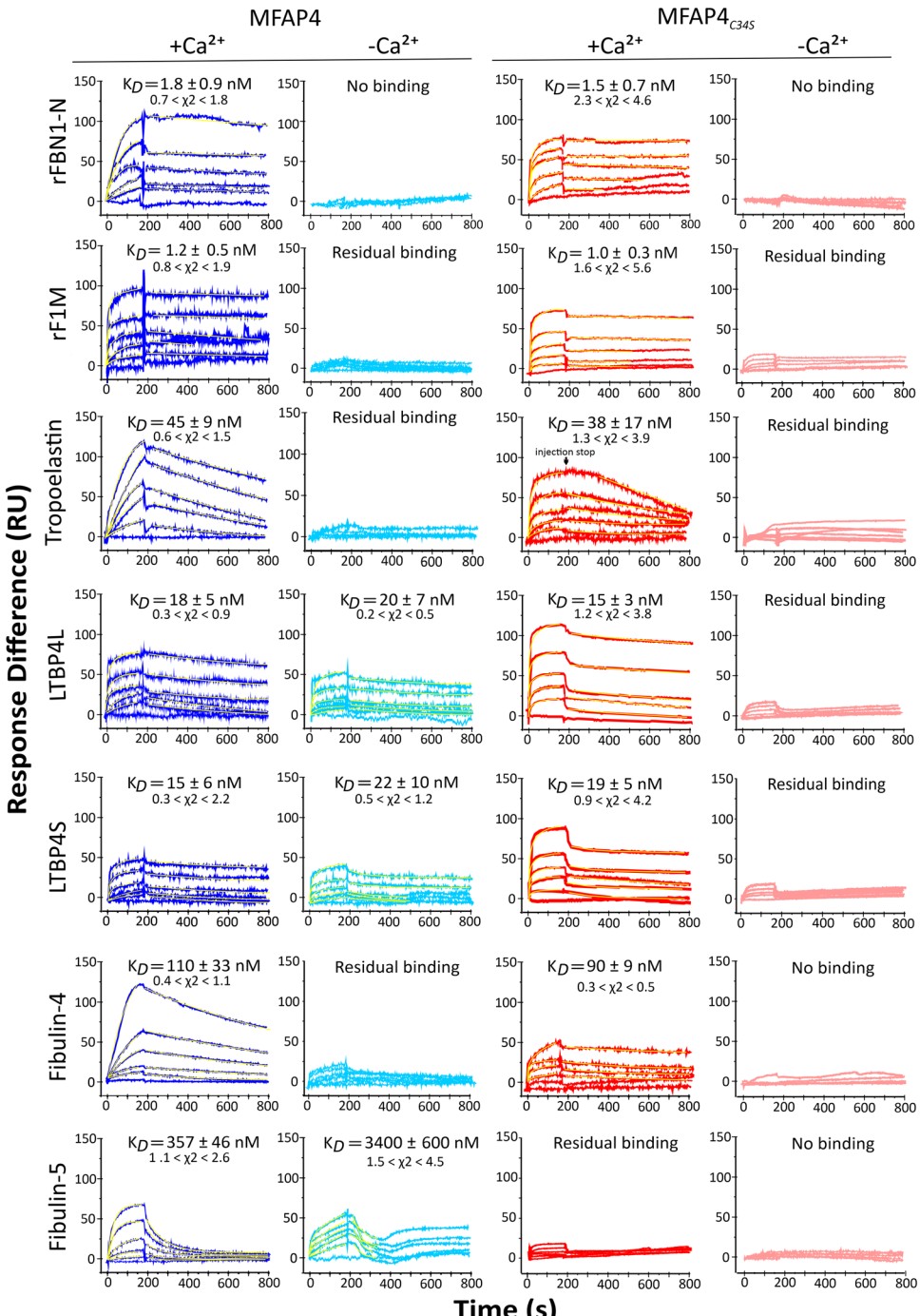

**Fig. 4 | MFAP4 and MFAP4$_{C34S}$ interactions with elastic fibre-associated extra-cellular matrix proteins.** The interactions were tested by SPR with MFAP4, and MFAP4$_{C34S}$ immobilised on the sensor surface either in the presence (+Ca$^{2+}$) or absence (-Ca$^{2+}$; EDTA treatment followed by TBS dialysis) of calcium. Full-length proteins and subfragments (as indicated) were used as analytes in the soluble phase either in TBS containing Ca$^{2+}$ (+Ca$^{2+}$) or in TBS without calcium (-Ca$^{2+}$). Each SPR profile corresponds to a different protein concentration (From top down: 100, 50, 20, 10, 5 and 0 µg/mL). The fittings are overlaid on the SPR traces in yellow. The χ2 values range (minimum-maximum) of the fittings are indicated. Additional inter-actions and negative controls are shown in Supplementary Table 1. Source data are provided as a Source Data file.

disulphide bonds are not necessary for protein interactions requiring octameric MFAP4, but they do affect those interactions that only require the MFAP4 tetramer.

Since the RGD motif required for cell interaction is located close to C34, we tested attachment of human dermal fibroblasts to MFAP4 and MFAP4$_{C34S}$ (Fig. 5A). MFAP4$_{C34S}$ promoted fibroblast attachment similarly to MFAP4, demonstrating that the C34-mediated inter-molecular disulphide bonds between MFAP4 monomers are not

required for the cells to interact with the RGD site. Because MFAP4 interacts with fibrillin-1 with very high affinity at the N-terminus, we analysed complexes of the N-terminal half as well as full-length recombinant fibrillin-1 with MFAP4 by AFM (Fig. 5B). Particles consistent with the diameter of the MFAP4 octamer occurred frequently at the end of rFBN1-N and full-length fibrillin-1. In some instances, longer assemblies of full-length fibrillin-1 were observed with an MFAP4 spacing of about one fibrillin molecule length (~130 nm). This data

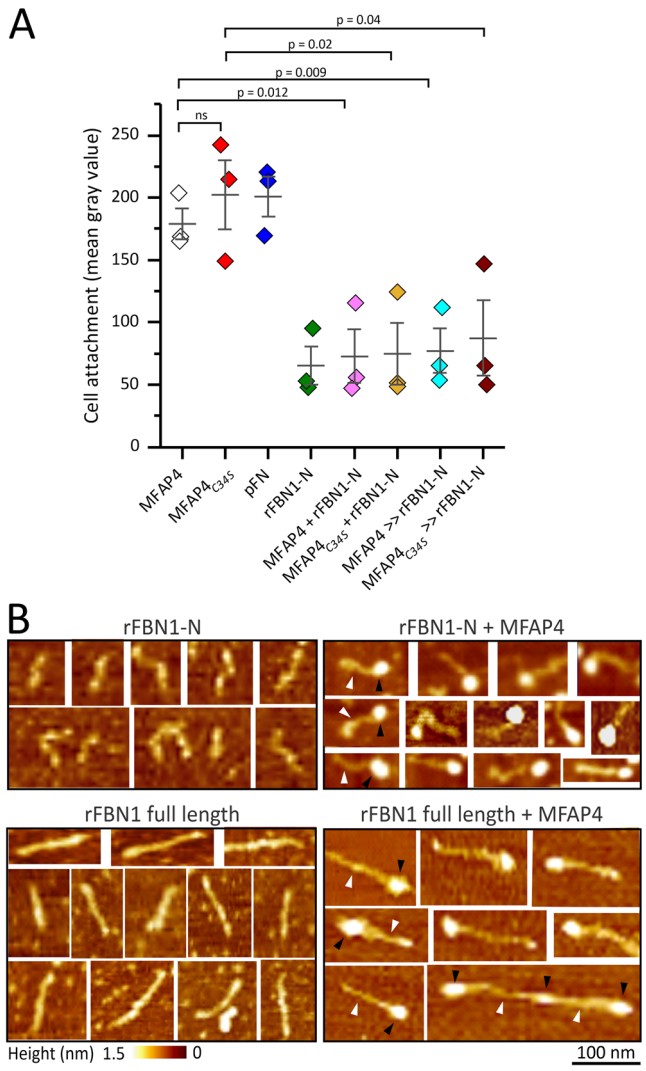

**Fig. 5 | MFAP4 and MFAP4$_{C34S}$ interactions with fibroblasts and visualisation of MFAP4-fibrillin-1 complexes. A** Cell attachment of human dermal fibroblasts to MFAP4, MFAP4$_{C34S}$, rFBN1-N, and plasma fibronectin (pFN) as control. The proteins were either coated alone as indicated or were first mixed (MFAP4/ MFAP4$_{C34S}$ + rFBN1-N) and then coated, or MFAP4/MFAP4$_{C34S}$ were first coated followed by incubation with rFBN1-N (MFAP4/MFAP4$_{C34S}$ » rFBN1-N). Data points represent mean values derived from $n = 3$ independent experiments, each measured with six technical replicates. Error bars represent the standard error of the mean. Statistical analysis was performed using the two-sided unpaired t-test. p-values are indicated for each comparison. Source data are provided as a Source Data file. **B** AFM visualisation of the recombinant N-terminal half of fibrillin-1 (rFBN1-N; top panel) or full-length recombinant fibrillin-1 (rFBN1; bottom panel) either alone (left panels) or mixed with MFAP4 (right panels). Some rFBN1-N and rFBN1 molecules are marked by white arrowheads, and some MFAP4 molecules are labelled with black arrowheads. The distance scale bar represents 100 nm for all images; the height scale bar represents 0-1.5 nm for all images.

suggested that MFAP4 interacts with the fibrillin-1 N-terminus close to the site of N-to-C fibrillin-1 self-interactions[7]. We further analysed whether interactions of MFAP4 and MFAP4$_{C34S}$ with cells were modulated upon fibrillin-1 binding. When the MFAP4 proteins were either pre-mixed with rFBN1-N before coating the wells, or when they were first coated and then incubated with rFBN1-N, adhesion to fibroblasts was abolished (Fig. 5A). This data shows that the RGD motif at the octamer poles become inaccessible for cell surface integrins upon fibrillin-1 binding, either through a competitive interaction mechanism or through steric hindrance.

## MFAP4 forms chains of octamers

In addition to isolated MFAP4 octamers, extended chain-like structures were visible in micrographs of MFAP4 vitrified in TBS/Ca$^{2+}$ (Fig. 2A). We investigated these higher-order assemblies by reanalysing the aligned dataset, extracting particles with a larger box size (~723 Å box-width) to capture partner octamers, 2D classification to classify particles into octamer chain classes and solitary octamer classes (Fig. 6A), followed by C1 point group symmetry refinement, first with a spherical mask (520 Å in diameter) and then a cylindrical mask, which included both the central octamer and the nearby partner (Fig. 6B). This scheme yielded a cryo-EM density consensus map with a central octamer and a partner octamer (9.07/15.92 Å resolution, masked/ unmasked FSC$_{half-map}$ = 0.143, Supplementary Fig. 10; local resolution was estimated with CryoRes[39], Supplementary Fig. 11; FSC estimation and angular distributions of central octamer, Supplementary Fig. 12, and partner octamer, Supplementary Fig. 13), suggesting that these chain-like assemblies are sufficiently flexible to be structurally disordered beyond two units (~266 Å). To further investigate the interface between these linked octamers, we utilised multi-body refinement within RELION 4-beta[40] to focus two individual maps on either the central octamer or its partner[41]. Multibody refinement yielded two maps with extensions reaching toward each other (Fig. 6C, D; Supplementary Movie 4). Principle component analysis shows that the primary and secondary modes of movement consist of the partner swinging ±30° relative to the central body (Fig. 6E, red and yellow partners). The tertiary mode of movement consists of the partner octamer twisting around an axis of connection between the two molecules (Fig. 6E, blue partner). We ascribe this connecting density between octamers, which was not resolved in the D2 map of a single octamer, to the sequence between Q21, the first residue after signal peptide cleavage, and C34, which is involved in the intermolecular disulphide bond within a single octamer. From these observations, we propose a pole-to-pole organisation of MFAP4 octamers interacting with each other through N-termini directed from the top and bottom of each octamer. Whether MFAP4 forms chains in situ and whether this may be regulated by RGD cell interactions is currently unknown.

## Discussion

MFAP4 is an extracellular protein that has been associated with elastic fibre formation and several diseases with defects in elastic fibres, including Marfan syndrome and chronic obstructive pulmonary disease[1,3,18]. To advance knowledge of how MFAP4 structurally and functionally relates to elastogenesis and the underpinning diseases, we determined the structure of human MFAP4 by cryo-electron microscopy and revealed important structure-function relationships in the context of multimerisation, calcium binding, ligand interaction, glycosylation, and cell interaction.

Previous studies have shown that MFAP4 and its bovine homologue MAGP-36 exist as disulphide-bonded dimers under denaturing conditions[10,27,35]. The existing paradigm is that these dimers assemble into higher-order multimers, albeit there is discrepancy on the type of assembled multimers, i.e., hexamers, octamers, or dodecamers[10,27]. Here, we provide evidence that C34 is sufficient and necessary for intermolecular disulphide bond formation of MFAP4 using the MFAP$_{C34S}$ mutant. Surprisingly, however, we found that the C34-mediated intermolecular disulphide bond is not required for higher-order assembly. For wild-type MFAP4 it is presently not known whether the intermolecular disulphide bond forms within the secretory pathway or possibly later after secretion. Our data favour a role of the C34-mediated intermolecular disulphide bond in stabilizing higher-order assembly, without being strictly required, as the hydrodynamic radius of the assembled complex decreases more for MFAP$_{C34S}$ than for MFAP4 with increasing NaCl concentration.

The 3.55 Å resolution cryo-EM structure of human MFAP4 in the presence of Ca$^{2+}$ unequivocally demonstrates an octamer structure

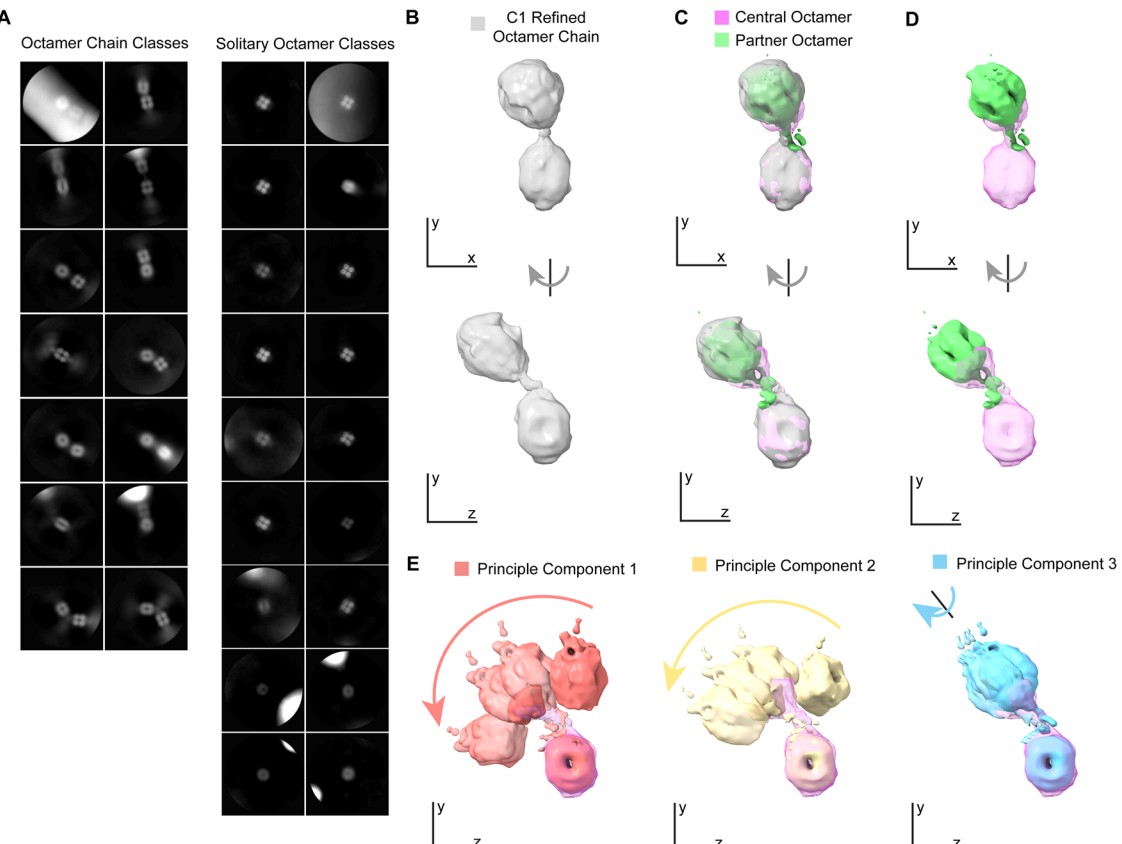

**Fig. 6 | MFAP4 octamer chains. A** 2D classification of MFAP4 particles in TBS/Ca$^{2+}$ reveals octamer-chain classes and solitary octamer classes (box size is 723.0 Å nm). **B** Particles from the selected octamer-chain 2D classes were used to generate and refine a map of two closely apposed MFAP4 octamers with connecting EM-density (grey). This reference map was used for masking and subsequent multibody refinement. **C** Maps of the central MFAP4 octamer (pink) and partner octamer (green) produced from multi-body refinement are presented superimposed on the consensus map of both bodies and (**D**) alone. **E** The first, third, eighth, and tenth multibody refinement maps are shown along the first three principal components of movement. The first (red) and second (yellow) principal components of movement describe a swinging movement of the partner relative to the central MFAP4 from left to right or right to left, respectively. The third (blue) principle of movement describes a twisting movement of the partner relative to the central octamer. These maps are shown superimposed on the central octamer map.

and thus solves the existing discrepancy in the field. Octamer formation of MFAP4 contrasts with the ability of other related FReD-containing proteins to multimerize. Fibrinogen C Domain Containing 1 protein (FIBCD1) forms tetramers (4M7H)[31], ficollin-2 forms trimers (2J3G)[42], and the FReD of angiopoietin-1 (Ang1) forms dimers (4EPU). A distinct conformation of loop A182-A206 in MFAP4 which distinguishes it from other FReD-containing proteins likely explains these differences as outlined below.

In the absence of Ca$^{2+}$, MFAP4 adopts a tetrameric conformation, which represents the identical top or bottom halves of the octamers. Each tetramer contains two sets of protomers linked as dimers by intermolecular disulphide bonds between C34. The fact that after removal of Ca$^{2+}$ tetramers remain stable supports a model of assembly where two MFAP4 protomers associate through intra-tetrameric interactions, predominantly through salt-bridge and hydrogen bonds. The data suggest that Ca$^{2+}$-binding is necessary for the conformation of loop A182-A206 and is stabilised by Ca$^{2+}$-binding to S186, T187, D191, and Q192. This conformation presumably is necessary for inter-tetrameric interactions, which are predominantly non-polar, non-bonded interchain interactions between V96-F93, L202-L202, S204-L202 apart from a single polar non-bonded interchain interaction between N231-A200.

In the presence of Ca$^{2+}$, we also observed formation of concatenated octamers forming a chain-like assembly with an approximate distance of 16 nm between octamers. This is likely based on the very high self-assembly affinity of octamers in the low nanomolar range. The octamers interact in a pole-to-pole arrangement, possibly through their exposed N-termini, which are not resolved by the cryo-EM map but must be situated near the top and bottom poles of MFAP4 octamers. This view is also supported by the fact that the tetramers observed in the absence of Ca$^{2+}$ never form any chain-like conformation. The preferential orientation of tetramers likely discourages interaction between the N-termini emanating from the centre of tetramers, perpendicular to the monodispersed single layer of vitrified MFAP4 without Ca$^{2+}$. If two tetramers are still able to interact with each other via their poles, it would result in an octamer that cannot further assemble because of the lack of required Ca$^{2+}$ for the inter-tetrameric interactions. Interestingly, the proposed pole interaction sites for chain formation are situated close to the RGD cell binding site close to the N-terminus, which interacts with integrin αvβ3 and possibly αvβ5 of elastogenic cells, facilitating cell migration and proliferation[22,30]. MFAP4 and MFAP4$_{C34S}$ interacted similarly with cells, but the capacity of both proteins' cell binding was significantly reduced upon interaction with the N-terminal region of fibrillin-1, suggesting that fibrillin-1 negatively regulates cell binding to MFAP4. Whether MFAP4 chain formation also controls cell interaction, whether it indeed occurs under physiological conditions, and how it is regulated remains to be investigated.

The interaction survey of MFAP4 with relevant elastogenic proteins revealed Ca$^{2+}$-dependent interactions with fibrillin-1, tropoelastin, fibulin-4, and fibulin-5, with the strongest affinity for fibrillin-1 in the low nanomolar range. Binding to fibrillin-1 and tropoelastin was

reported previously, albeit MFAP4 interaction with fibrillin-1 was not dependent on $Ca^{2+}$ in that study[10]. While binding to fibulin-4 and -5 was of moderate affinity, it is possible that MFAP4 is involved in the chaperone function that has been described for fibulin-4 converting folded LTBP4L to an extended conformation which can then align with microfibrils to attract tropoelastin[11]. We identified here that MFAP4 also interacts with an N-terminal region of LTBP4L and LTBP4S, but this interaction is not dependent on $Ca^{2+}$. Together, the data support the interpretation that the $Ca^{2+}$-dependent MFAP4 interactions likely require the octameric MFAP4 conformation, whereas the $Ca^{2+}$-independent interactions only require MFAP4 tetramers that are prevalent in the absence of $Ca^{2+}$. MFAP4 octamers could provide distinct surfaces that are unavailable or distorted in the absence of $Ca^{2+}$ as tetramers. $Ca^{2+}$ concentrations in the extracellular matrix are generally assumed to be in equilibrium with circulating blood in the range of 1-2 mM $Ca^{2+}$. Whether or not $Ca^{2+}$ concentrations in specific extracellular spaces, for example, in cell invaginations[43], can be regulated to a level that would allow MFAP4 tetramers to occur is currently not known. Also, the affinity of MFAP4 for $Ca^{2+}$ that determines the level of $Ca^{2+}$ saturation under physiological conditions remains to be determined. Intriguingly, $MFAP4_{C34S}$ bound to the tested proteins with nearly identical affinities in the presence of $Ca^{2+}$, with exception of fibulin-5 did not interact. Another difference in the interaction profile between MFAP4 and $MFAP4_{C34S}$ was that the latter interacted with LTBP4L and LTBP4S in a $Ca^{2+}$-dependent manner. The data suggest that in the absence of $Ca^{2+}$, the interactions of MFAP4 with fibulin-5 and the LTBP4 isoforms depend on stabilisation of the pole regions by inter-molecular disulphide bonds, but other interactions do not require this stabilisation.

Although FReD family members have different ligand specificities, ligand binding is mediated in part by the S1 pocket, corresponding to aa 187-251 of MFAP4[10]. Binding to tropoelastin and collagen type I has previously been investigated by mutating S203 and F241 within the S1 site of MFAP4[10]. Both F241A and F241W mutations strongly attenuated or abolished tropoelastin and collagen type I binding. A hydrophobic surface is formed near the interfaces between tetrameric halves of the octamer by L227, Y229, F241, and Y242; this surface, or at least F241, could represent a potential tropoelastin/collagen type I binding site. Interestingly, the S203Y, but not the S203A mutation, also decreased tropoelastin and collagen type I binding[10]. The cryo-EM map shows that S203 is not exposed in the assembled MFAP4 octamer complex; instead, S203 is tucked within the interface between tetrameric halves of the octamer. We suspect that the S203Y mutation disrupts this interface, possibly by preventing proper folding of the nearby region, which is stabilised by a disulphide bond between C199-C212, while S203A is tolerated. Misfolding in this region could potentially prevent C199-C212 disulphide bond formation and/or interfacing between MFAP4 tetramers, thus preventing octamer formation. This is in line with tropoelastin requiring the $Ca^{2+}$-saturated MFAP4 octamer for binding but fails to bind to the $Ca^{2+}$-depleted tetramers.

N-linked glycans emanating from N87 and N137 on the surface of MFAP4 octamers suggest that these moieties might influence ligand binding or provide additional ligand binding sites. In the D2 point group symmetry cryo-EM map, both N87 and N137 appear to be simultaneously glycosylated; although, the glycan at N137 is more flexible than N87, which has a stronger signal-to-noise ratio within the cryo-EM map. Interestingly, N87 is closer to the MFAP4 octamer centre of mass and positioned near the central cavity. It is possible that the glycan at N87 is partially stabilised by this surrounding environment while the glycan bound to N137 can flex within the surrounding solvent. MFAP4 N-linked glycosylation patterns are elevated and more diverse in aneurysmal ascending aortic tissues of patients with Marfan syndrome[26]. Altered MFAP4 glycosylation could potentially affect binding to elastogenic proteins, including fibrillin-1 or LTBP4 isoforms, possibly affecting formation and maintenance of microfibrils and elastic fibres, which are compromised in Marfan syndrome. Since the

N-linked glycans in MFAP4 are positioned far from F241, it is unlikely that N-glycan alterations would interfere with tropoelastin/collagen type I binding to this site. Given the exposed position on the MFAP4 octamer, the N-linked glycans could also directly promote interactions with elastogenic proteins, possibly altering the stability of the elastic fibre system in the ascending aorta. Conversely, C-mannosylation of W235 is likely to play a structural role, given the buried position of W235 within MFAP4. Although we did not observe mannosylation of W235, the C-mannosylation target motif (WXXW) is present, and previous reports clearly indicate that W235 mannosylation affects MFAP4 secretion[29]. This difference could arise from the different cell lines used in our study (HEK293) and others (HT1080 human fibrosarcoma). In the event of W235 mannosylation, we would expect a conformational change to accommodate the additional mannose monosaccharide within this buried region.

In the last stages of writing this manuscript, an X-ray crystal structure of human MFAP4 in complex with a Fab fragment was made available in the Protein Data Bank (PDB:7ZMK) by Laursen et al. This structure remains undescribed in the literature and is not linked to a manuscript available on a preprint server. Therefore, we cannot discuss differences from a biochemical and functional perspective, but we discuss here the structural differences compared to our cryo-EM structure. Aligned individual monomers exhibit an excellent fit for the core of the molecule, with a root-mean-squared deviation (RMSD) of 0.79 Å (198 residue pairs) but a more modest fit for all 220 residues (RMSD: 2.1 Å). The largest differences are in the loops F188 - A206 and R210 – N216 that form the tetramer-tetramer contact. When the top tetramer of 7ZMK is aligned with the tetramer described in our study, it results in an RMSD of 3.56 Å. The bottom tetramers, however, have an RMSD of 19.6 Å. This is because the octamer described in our study has a more extended conformation. The 7ZMK structure has its lower tetramer rotated so that portions of the lower monomers nestle between two upper monomers. Each monomer then forms contacts with 4 other monomers in the 7ZMK structure instead of only 3 others in our model. In the absence of any information about the biochemical treatment of this MFAP4-Fab complex, we speculate that 7ZMK possibly represents an inactive state of MFAP4 since its ligand binding sites are less exposed to solvent.

In conclusion, we present a 3.55 Å resolution structure of human MFAP4, which assembles into octamers and, under certain conditions, into extended chain-like structures. We clarified the role of $Ca^{2+}$ in the structure and provided extensive structure-function analyses in terms of interaction with elastogenic proteins and cells. This knowledge should provide a solid basis for understanding MFAP4's role not only in elastogenesis but also in the various diseases it is associated with.

## Methods

### Protein preparation and purification

For the production recombinant full-length human MFAP4, a double-stranded DNA coding for full-length MFAP4, including adjacent DNA sequences for cloning, was commercially synthesised (g-block; Integrated DNA Technologies) (Supplementary Table 2). The synthetic DNA was cloned into the *NheI* × *NotI* restricted pCEP4 mammalian expression vector (Invitrogen) using the Gibson Assembly Cloning Kit as instructed by the manufacturer (New England BioLabs). The resulting vector was termed pCEP4-MFAP4 coding for the amino acid sequence $M^1KAL...IRRA^{255}$. An additional hexahistidine tag at the C-terminal end facilitated purification of the recombinant MFAP4. To produce $MFAP4_{C34S}$, the codon for C34 in pCEP4-MFAP4 was modified from TGC to TCG using the QuickChange Site-Directed Mutagenesis kit as instructed by the manufacturer (Agilent) with the oligonucleotides shown in Supplementary Table 2. The entire insert in pCEP4-MFAP4 and the sequence coding for the C34S mutation in pCEP4-$MFAP4_{C34S}$ was validated by Sanger DNA sequencing. Transfection of HEK 239 EBNA cells (Invitrogen; catalogue #R620-07), selection,

production of conditioned medium containing recombinant MFAP4 or MFAP4$_{C34S}$, and purification by nickel chelating chromatography followed previously established procedures[44,45]. After purification, the proteins were dialysed and stored in 50 mM Tris-HCl, pH 7.4, 150 mM NaCl, 2 mM CaCl$_2$ (TBS/Ca$^{2+}$). Some protein samples have been treated with EDTA (Fisher Scientific, catalogue #BP118-500) or EGTA (Bio Basic, catalogue #67-42-5) for analysis in the absence of Ca$^{2+}$.

For SDS-PAGE analysis, proteins were electrophoresed in the presence (50 mM) and in the absence of DTT (Roche, catalogue #10708984001) in SDS sample buffer on 10% gels. For titration experiments with DTT, MFAP4 was incubated with 0-50 mM DTT under physiological conditions (TBS/Ca$^{2+}$) for 30 min at ambient temperature. The solution was supplemented with 100 mM iodoacetamide (Bio-Rad, catalogue #163-2109) and incubated in the dark for 1 h. Samples were then electrophoresed (without additional DTT) on 10% SDS gels and stained with Coomassie Blue. Original uncropped and unprocessed gels are provided in Source Data file.

Production and characterisation of recombinant fibulin-3, fibulin-4 and fibulin-5[28], LTBP4L and LTBP4S[11], full-length fibrillin-1[46], and the fibrillin-1 subfragments rFBN1-N and rFBN1-C[47], and rF1M[44], were previously described in detail. Recombinant human tropoelastin was purchased from Elastagen, and purified human plasma fibronectin was obtained from Millipore.

## Dynamic light scattering

Particle sizes of MFAP4 and MFAP4$_{C34S}$ were assessed by DLS (DynaProMolecular-Sizing Instrument, Protein Solutions, Wyatt Technology). Acquisitions of 5 s or 10 s readings were recorded for each measurement and averaged. Samples from different MFAP4 and MFAP4$_{C34S}$ purification batches were analysed in quadruplicates, and the results were averaged. Samples were used in TBS/Ca$^{2+}$ (2 mM) with or without treatment of 5 mM EDTA or 5 mM EGTA for 20 min as indicated. Some samples were analysed under denaturing conditions in the presence of 1% SDS in TBS either with or without reducing DTT (50 mM) or titrated with 0-2 M NaCl. For divalent cation titrations, MFAP4 and MFAP4$_{C34S}$ were first Ca$^{2+}$-depleted with 5 mM EGTA (20 min) and then supplemented with divalent cations at physiologically relevant concentrations (2 mM Ca$^{2+}$ or Mg$^{2+}$; 15 μM Zn$^{2+}$ or Cu$^{2+}$; 0.15 μM Mn$^{2+}$). Molecular mass analyses of MFAP4 and MFAP4$_{C34S}$ from measured hydrodynamic radii were performed by applying the globular (compact) model for the conformational protein shape using calibration curves developed from protein standards of known molecular masses provided by the manufacturer (DynaPro).

## Transmission electron microscopy and atomic force microscopy

Negative stain TEM and AFM were performed with MFAP4 and MFAP4$_{C34S}$ either in the Ca$^{2+}$-form (TBS/Ca$^{2+}$) or in the Ca$^{2+}$-depleted form (5 mM EDTA-treated) as indicated. Negative-stain TEM was performed on a FEI Tecnai 12 Spirit 120 kV TEM microscope, and AFM imaging was performed with a Multimode HR 8 instrument (Bruker) in tapping mode in air. For the AFM analyses, MFAP4 and MFAP4$_{C34S}$ were absorbed on freshly cleaved Muscovite grade 1 mica surfaces for a few min and quickly dried with pressurized air. The AFM probes (TESPA-V2; Bruker) had a nominal tip radius of 2 nm. Quantification of size, shape, and abundance of MFAP4 and MFAP4$_{C34S}$ structures was performed by post-processing analysis of the AFM height images with molecules selected from multiple images and experiments using the instrument software (NanoScope Analysis 2.0). Image processing included surface flattening as well as adjustments of the Z-axis scale and offset, brightness, and contrast. Particle shape and size quantification was performed to ensure adequate reproducibility of several protein preparations with a minimum of 300 particles per protein preparation and condition. Molecular structures of more complex shapes or aggregates were excluded from the analysis. Particle heights were quantified with the 2D section profile feature of the AFM software

using traces drawn in the centre of each particle to determine the maximum z-axis (height) value. For some experiments, MFAP4 was mixed with the N-terminal half (rFBN1-N) or full-length recombinant fibrillin-1 (1:1 protein mass concentration; 50 μg/mL of each component) and then imaged by AFM.

## Cryo-transmission electron microscopy and atomic modelling

MFAP4 was vitrified on CF-2/1-3Cu grids (Protochips, 20 nm carbon, 2.0 μm hole diameter, 1.0 μm hole spacing, 300 mesh copper grid, catalogue #CF-2/1-3CU-50) as 3.5 μL of sample (0.551 mg/mL; either in the Ca$^{2+}$-form directly from the TBS/Ca$^{2+}$ stock, or after depletion of Ca$^{2+}$ by 5 mM EDTA treatment), following 1.5 s of blotting with blotforce 10 using a Vitrobot at 4 °C and 90% humidity. Images were collected on a FEI Titan Krios TEM operating at 300 kV at a nominal magnification of 105,000 with beam-shift using SerialEM. Images were recorded on a Gatan K3 camera with pixel size of 0.855 Å/px. Images were dose-fractionated as 40 frames over a total exposure time of 3.356 s with 0.0839 s per frame and a frame dose of 2.00 electrons per Å$^2$. The defocus range was −1.0 to −2.5 μm under focus. 6,751 images were collected.

The raw frames were motion-corrected, summed, and dose-weighted with MotionCor2[48]. Contrast transfer function (CTF) parameter estimation was carried out on the non-dose-weighted, motion-corrected images with CTFFIND4[49]. Particles of MFAP4 with Ca$^{2+}$ were manually picked using RELION4-beta for the initial 2D model used to train the neural network Topaz for automatic particle picking; an initial number of 878,597 putative particles were identified within 6,714 micrographs (Supplementary Table 3). After 2D classification 458,868 particles were extracted with a box size of 320 px (27.36 nm), rescaled to 80 px, and used for initial 3D reconstruction. After 3D refinement and further classification, 444,005 particles, with a box size of 320 px (27.36 nm), contributed to a 3.18/3.55 Å (masked/unmasked) resolution structure of the entire MFAP4 octamer, reconstructed with D2 symmetry (EMDB ID: EMD-42394). B-factor map sharpening was performed with RELION4-beta. Resolution was estimated from the FSC at 0.143 from a masked or unmasked reference with Phenix v1.20.1-4487[34] (mask smoothing radius 6.58 Å). An overview of the single particle analysis workflow for MFAP4 with Ca$^{2+}$ is presented in Supplementary Fig. 14.

These 444,005 particles were re-extracted with a larger box size of 846 px (72.33 nm) and rescaled to 282 px for further processing to reveal octamer partners. 2D image alignment and classification were used to separate octamer chain class particles from solitary octamer class particles based on the presence of a nearby partner octamer or not. 173,879 octamer chain class particles were selected, and an initial map was generated with RELION4-beta's 3D initial model function from these particle images. This initial map was refined with a spherical mask and subsequently a cylindrical mask, which was positioned along a vector drawn between the two octamer's centres of mass. 173879 particles, with a box size of 282 px (72.33 nm) contributed to a 9.07/15.92 Å (masked/unmasked) resolution consensus map of adjacent MFAP4 octamers, which we refer to as an octamer chain, reconstructed with C1 point group symmetry. An overview of the single particle analysis workflow for chains of MFAP4 with Ca$^{2+}$ is presented in Supplementary Fig. 15. This refined consensus map was segmented in ChimeraX to produce separate maps of the central and partner octamer. These separate maps were used to generate masks with RELION 4-beta. These were applied to the refined consensus map using Sparx[50] to produce references for multibody refinement[50] Masks for multibody refinement were finally produced using RELION4-beta from the references used for the multibody refinement. Multibody refinement calculations were done using RELION4-beta's 3D multi-body function[41]. B-factor map sharpening was performed with RELION4-beta. Resolution was estimated from the FSC at 0.143 from a masked or unmasked reference with Phenix v1.20.1-4487[34] (mask smoothing radius 18.0 Å).

Particles of MFAP4 without $Ca^{2+}$ were manually picked for the initial 2D model used for template matching and automatically picked with Topaz within RELION4-beta[40]. An initial number of 914,002 putative particles were identified in 5,800 micrographs. After 3D refinement and further classification, 578,476 particles, with a box size of 320 px (27.36 nm) and rescaled to 160 px, contributed to a 4.46/5.26 Å (masked/unmasked) resolution map of a MFAP4 tetramer (EMDB ID: EMD-42398), reconstructed with C2 symmetry. B-factor map sharpening was performed with RELION4-beta. Resolution was estimated from the FSC at 0.143 from a masked or unmasked reference with Phenix v1.20.1-4487[34] (mask smoothing radius 30.0 Å). An overview of the single particle analysis workflow for MFAP4 without $Ca^{2+}$ is presented in Supplementary Fig. 16.

Local resolution of MFAP4 with $Ca^{2+}$ was estimated using ResMap[38] with one of the two independently refined half-maps before B-factor sharpening along with the mask volume used for B-factor sharpening as input for ResMap. The local resolution map from ResMap was mapped onto either a half-map or the RELION4-beta B-factor sharpened postprocessed map at low and high contour levels using ChimeraX. Local resolution estimation of MFAP4 without $Ca^{2+}$ and the octamer chain (consensus and multi-body refined maps) with ResMap produced unreliable local resolution estimates, which ascribed resolution estimates of <5 Å to map features without secondary structure. Instead, the local resolution was estimated using CryoNet"s trained neural network[39,51]. Local resolution maps were generated from either one of the two independently refined half-maps or the B-factor sharpened postprocessed map of MFAP4 without $Ca^{2+}$, as well as the consensus and multibody refined maps of the MFAP4 octamer chain. ChimeraX was used to project these local resolution maps onto the maps from which they were generated[52]. A starting model of human MFAP4 (Supplementary Data 1) was predicted with AlphaFold Colab (https://colab.research.google.com/github/deepmind/alphafold/blob/main/notebooks/AlphaFold.ipynb), a simplified version of AlphaFold v2.3.1[53]. This starting model was initially docked within the presented cryo-EM map of MFAP4 with $Ca^{2+}$ using rigid body fitting in Chimera[54]. This predicted model was refined manually in Coot (v0.9.8.1)[33], then automatically refined within Phenix (v1.20.1-4487)[34]. Iterative manual and automatic refinement steps were performed with Coot and Phenix to obtain the atomic model (PDB ID: 8UN7). A map-to-model fixed radius cross-correlation score was calculated according to Jiang and Brünger, 1994, with Phenix during automatic map refinement[55]. Ligands within the model are displayed as ball-and-stick structures within the cryo-EM map density corresponding to both ligand and interacting residues at the same sigma level (Supplementary Fig. 17).

## Protein and cell binding assays

Surface plasmon resonance (SPR) spectroscopy was used to study binding characteristics and kinetics of MFAP4 and $MFAP4_{C34S}$ interaction with itself and with elastic fibre-associated proteins referenced above (fibrillin-1 sub-fragments rFBN1-N (N-terminal half), rFBN1-C (C-terminal half), rF1M (centre region), tropoelastin, LTBP4L (long isoform), LTBP4S (short isoform), LTBP4L/S N-terminal and C-terminal halves, fibulin-3, fibulin-4, fibulin-4, and fibronectin). The experiments were performed with a Biacore X instrument using CM5 sensor chips (Cytiva, catalogue #29149604). Ligand protein MFAP4 or $MFAP4_{C34S}$ was covalently immobilised onto one of the two channels of the sensor chip by amine coupling using the standard protocol supplied by the manufacturer with immobilisation levels in the range of 300-500 resonance units (RU). Different sensor chips were generated for $Ca^{2+}$-loaded and $Ca^{2+}$-depleted MFAP4 or $MFAP4_{C34S}$ as follows. MFAP4 or $MFAP4_{C34S}$ in TBS/$Ca^{2+}$ was either diluted in immobilisation buffer (10 mM sodium acetate, pH 4) to generate the $Ca^{2+}$-form protein sensor chips or treated with 5 mM EDTA for 20 min followed by dialysis in TBS to generate the $Ca^{2+}$-free protein sensor chips. The second channel of

the sensor chip was left blank (no protein coated), which served as control for analyte binding specificity after inactivation of the reactive amine groups. Kinetic analyses were performed in TBS/$Ca^{2+}$ for interactions in the $Ca^{2+}$-form, or in TBS buffer for interactions without $Ca^{2+}$. Binding experiments were performed by injecting 30 μL of analyte at concentrations of 100, 50, 20, 10, 5, and 0 μg/mL at a 10 μL/min flow rate. Molecular association was monitored for 180 s and dissociation for 600 s under identical flow conditions. For MFAP4 and $MFAP4_{C34S}$ interaction assays in the $Ca^{2+}$-form, the analyte was used in TBS/$Ca^{2+}$. For $Ca^{2+}$-free MFAP4 and $MFAP4_{C34S}$ self-interactions, samples in TBS/$Ca^{2+}$ were first treated with EDTA or EGTA (5 mM) for 20 min and then dialysed against TBS. For the $Ca^{2+}$-free interactions with elastogenic proteins, the analyte stored in TBS/$Ca^{2+}$ was dialysed against TBS to remove $Ca^{2+}$ from the buffer. This procedure ensured that the analyte protein was $Ca^{2+}$-saturated during the interaction with $Ca^{2+}$-free MFAP4 or $MFAP4_{C34S}$. Some binding assays were performed in running buffer TBS, including 2 mM $MgCl_2$ (TBS/$Mg^{2+}$). For these experiments, MFAP4 or $MFAP4_{C34S}$ were treated with 5 mM EGTA followed by dialyses against TBS/$Mg^{2+}$. The association-dissociation sensorgrams were fitted with the BIAevaluation software (Cytiva) using the "Fit Kinetics Separate" tool with 1:1 (Langmuir) binding model. The dissociation constants $K_D$ were calculated as the average of individual $K_D$ values determined for each of the five analyte concentrations tested (100, 50, 20, 10, and 5 μg/mL). The average $K_D$ and standard deviation are indicated on each sensorgram. The $\chi 2$ values describing the goodness of the fit and, hence, the quality of the data are indicated in the sensorgrams. The molecular masses of the analyte proteins used for the calculation of the molar concentrations were calculated from the protein amino acid sequence (theoretical) for monomers as well as octamers/tetramers as applicable. Some sensorgrams were corrected at the beginning and the end of the analyte injection for spike signals and bulk shift effects that can occur due to the short delay between the readings of the two flow channels and refractive index changes caused by the analyte solution. Offsets in Y-axis values at the injection time (0 s) were corrected to 0.

Cell attachment was tested using primary human dermal fibroblasts (local ethics board approval PED-06-054) following established procedures[56]. Briefly, $Ca^{2+}$-saturated MFAP4, $MFAP4_{C34S}$, fibrillin-1 fragment rFBN1-N (N-terminal half of fibrillin-1 ending prior to the RGD-containing TB4 domain[47]), and plasma fibronectin were coated at 50 μg/mL in TBS at 4 °C overnight onto 96-well Maxisorp plates (Nunc). In some cases, 50 μg/mL MFAP4 or $MFAP4_{C34S}$ was preincubated with 50 μg/mL rFBN1-N for 30 min at 22 °C, or alternatively, first coated onto the wells at 50 μg/mL and then additionally incubated for 1 h with rFBN1-N (50 μg/mL). Wells were washed with TBS and blocked with 10 mg/mL denatured bovine serum albumin. Near-confluent layers of fibroblasts were briefly treated with trypsin (3 min), counted, and seeded at 25,000 cells/well in 6 technical replicates for 1 h at 37 °C. The wells were extensively washed, fixed with 5% (w/v) glutaraldehyde in PBS (20 min at 22 °C), washed again, and stained with 0.1% (w/v) crystal violet (Sigma Aldrich). After extensive washing with TBS, the stained wells were photographed, and the mean grey pixel density was quantified by ImageJ[57].

## Reporting summary

Further information on research design is available in the Nature Portfolio Reporting Summary linked to this article.

# Data availability

Data generated in this study were deposited in the Electron Microscopy Data Bank under accession codes EMD-42394 and EMD-42398 and the Protein Data Bank under accession code 8UN7. The PDB code of the previously published structure used in this study is 7ZMK. Custom scripts are available on Zenodo (https://zenodo.org/records/10833841; https://zenodo.org/records/10988630; https://zenodo.org/

records/10834516). The model of human MFAP4 predicted with AlphaFold Colab is provided in Supplementary Data 1. All other data are available in the article and the Supplementary Information file. All source raw data are provided as a Source Data file. Source data are provided with this paper.

## Code availability

Code used to analyse maps and models can be accessed under the following DOIs: https://zenodo.org/records/10988630 (cylindrical-mask-making), https://zenodo.org/records/10833841 (pdb_at_dist) and https://zenodo.org/records/10834516 (mask_from_model).

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

## Acknowledgements

This study was supported by the Natural Sciences and Engineering Research Council of Canada (RGPIN-2022-05045 to DPR and RGPIN-2020-04837 to MS), the Genetic Aortic Disorders Association Canada (Gada Canada Aortic Research Grant Award 2024 to DPR), the Canadian Institutes of Health Research (PJT-186194 to MS; Postdoctoral Fellowship MFE-187851 to MRW). We thank Dr. Hojatollah Vali for assisting with negative stain TEM at the initial stage of the project and Dr. Kaustuv Basu for assistance with cryo-EM data collection. We also thank Dr. Marc McKee for access to AFM and Dr. Mari Kaartinen for access to DLS.

## Author contributions

Reinhardt D.P. and Strauss M. conceived the idea and designed the experiments, Wozny M.R., Nelea V., Siddiqui I.F.S. and Wanga S. designed and performed experiments, Wozny M.R. and Nelea V. wrote the manuscript, Reinhardt D.P., Strauss M., and de Waard V. troubleshot experiments, and Wozny M.R., Nelea V., Reinhardt D.P., and Strauss M. edited and finalised the original and revised manuscript. All authors contributed to the article and approved the submitted version.

## Competing interests

The authors have no competing interests.
