## [Peer Review File · Nature Communications]

Microfibril-associated glycoprotein 4 forms octamers that mediate interactions with elastogenic proteins and cellsREVIEWER COMMENTS

Reviewer #1 (Remarks to the Author):

This work presents a thorough analysis of the structure of microfibrillar-associated protein 4 (MFAP4). Combining experimental results from dynamic light scattering, atomic force microscopy, and cryo-electron microscopy with modelling tools, the authors conclude that MFAP4 assembles as octamers in the presence of calcium ions, but the octamers dissociate into tetramers in calcium-depleted conditions. A mutant MFAP4(C34S) was also studied to analyze the effect of disulfide bonds on the protein structure and stability. The binding of MFAP4 and MFAP4(C34S) to elastogenic proteins and cells (human dermal fibroblasts) were studied to understand how the structure of MFAP4 and the conditions (presence or absence of calcium ions) affect the interactions with elastogenic proteins and cells. The manuscript presents a rigorous experimental work and a detailed structural analysis that are relevant to better understand the function of MFP4 in elastogenesis, and it is also an important base for further research on the role of MFP4 in different diseases. The work is of interest for the readers of Nature Communications, but a minor revision is needed:

- 1) Supplementary Figures S2 and S3 are not referred to or commented in the main manuscript.
- 2) Figures 3B and 3C: it should be explained in the figure caption that each curve corresponds to a different protein concentration.
- 3) Figure 3G presents results from gel filtration chromatography, where 3 peaks are observed. The authors should elaborate the description in lines 250-254 a bit more and explain what oligomers are associated to each peak.
- 4) Some of the SPR sensograms in Figure 4 show a decrease with time, suggesting detachment of bound proteins (for instance, tropoelastin, fibulin-4 and fibulin-5 interacting with MFAP4 in the presence of calcium ions). Could the authors comment on that?
- 5) In lines 412-415 the authors state: "The data suggest that the interactions of MFAP4 with fibulin-5 and with the LTBP4 isoforms depend on stabilisation of the pole regions by inter-molecular disulfide bonds, but other interactions do not require this stabilisation." However, Figure 4 shows that MFAP4(C34S) can also bind LTBP4 isoforms in the presence of calcium ions, in contradiction with the previous statement.

Reviewer #2 (Remarks to the Author):

This manuscript describes the 3.55 Angstrom cryoEM structure of the octameric MFAP4, an extracellular protein linked to disease and pathological changes such as fibrosis. From their structure, the authors identify a number of new, interesting features; this includes a role for divalent cation binding in the

formation of the MFAP4 octamer, coordinated by residues around Asp 191. They also show that cysteine residue 34 is important for the formation of an intermolecular disulfide bond but which is not required for multimerization of MFAP4. They also show that some protein interactions require divalent cations. The cryoEM data and resulting structure appears to be of high quality, but I had some queries and comments about the manuscript that I hope the authors can address.

The cryoEM map enabled the authors to resolve the positions of residues Q37-A255 but the amino acid residues preceding Q37 were not observed in the cryoEM map. Despite this, the loop at the N-terminus is modelled. Is it appropriate to model this loop and the disulfide bond if the density isn't observed for this region in the map? If the disulfide bond at C34 has formed why are residues 34-37 disordered and not constrained?

Is the cryoEM data deposited in EMDB/EMPIAR, I didn't spot any accession numbers?

It is indicated that "relatively low levels of DTT (~3 mM) were required to reduce MFAP4 to monomers which supported a surface-located position for the intermolecular C34 disulfide bond". 3mM DTT seem a high concentration when compared to the likely protein concentration. How does 3mM compare to the concentration of MFAP4 in this experiment and would this concentration of DTT be expected to reduce inter vs intramolecular disulfide bonds?

The link to calcium is made as other FReD proteins bind calcium, and EDTA treatment of MFAP4 abolishes some interactions and decreases the size of the octamer. Why was EDTA used to chelate metal ions rather than the more calcium specific EGTA? Can the authors rule out the contribution of a different divalent cation, or show that calcium is indeed the bound metal ion in the cryoEM density? Perhaps titrating back in calcium specifically to monitor octamer formation and binding. A more elegant experiment might be to mutate the Aspartate residues that appear to coordinate the metal ion to see if this abolishes both octamer formation and metal-ion binding.

Following this point, metal ion binding appears to be a requirement for interaction with some proteins shown with SPR. It is not clear from the methods whether gel filtration was performed on samples prior to SPR and if the preparation of MFAP4 for both the WT and cys mutant is a mixture of oligomeric states (as suggested in the gel filtration analysis in Fig 3G). I don't think it is possible then to say that only the octameric form binds given the data presented, metal-ion binding is certainly required for the interaction and metal-ion binding stabilises the oligomers but decoupling these factors requires more subtlety.

On a related point, generally from the methods it is very difficult to determine how the protein was purified prior to cryoEM, SPR or DLS. Was gel filtration performed as a final purification step after affinity chromatography and what was the oligomeric state of MFAP4 in these experiments, ie purified octamer or a heterogeneous mixture as shown in Fig 3G prior to gel filtration in the presence of metal ions? For example, the DLS methods state "Samples of different MFAP4 and MFAP4C34S preparations in various buffer conditions, either gel filtrated or not, or treated with EDTA were analysed in quadruplicates and the results were averaged." The details of whether gel filtered protein or what buffer used is not in the figure legends so this is not clear.

Interestingly in Fig 1F, increasing NaCl concentration causes the hydrodynamic radius of MFAP4C34S to

reduce to a similar size as the calcium absent form, which is thought to be a tetramer (Fig3A). Why does changing salt concentration not cause the wildtype MFAP4 to dissociate to a tetramer if disulfide bonding does not stabilise the octameric form?

For figure 3, the TEM of the non-calcium bound structure does not show definitively that it is a tetramer due to the defined orientation and lack of side views. It would be nice to see some 2D classification of the side views, not just the top views, to show it is a tetramer. Were side-views present in negative stained images, and were attempts made to try different cryoEM grids, add a carbon support film or collect tilted data to optimise the conditions?

For figure 4, all SPR traces should include a key with the different concentration of analyte used (information also not in legend in fig 3). The methods indicate that a 1:1 Langmuir model was used for kinetic fitting, and this fit should be overlaid onto all kinetic traces along with chi-squared values for these fits (even if this is included in the supplementary material). Certainly, for some SPR traces they do not visually appear to represent a 1:1 binding fit and there are some unusual features in these data, for example MFAP4C34S with Ca²⁺ binding to tropoelastin it is hard to distinguish the association/dissociation phases.

For figure 5, there is very little detail about the data processing of the higher order oligomeric assemblies. The authors should include 3D maps of the oligomeric reconstructions and at least 2D classes to show the classification of the larger repeats.

Minor points:

State the box size or scale for the 2D classes in Figure 2.

In Figure 3, the panels should flow in alphabetical order, currently it is ABCGFDE.

Detailed responses to reviewer comments

We respectfully thank both the reviewers and the editor for the pertinent and fruitful comments, questions, and suggestions. They have led to significant improvements in the manuscript. Please find our point-to-point answers below. We reproduced all reviewers' comments in black font and provided our answers in blue font. All line numbers refer to the revised version of the manuscript.

REVIEWER #1:

This work presents a thorough analysis of the structure of microfibrillar-associated protein 4 (MFAP4). Combining experimental results from dynamic light scattering, atomic force microscopy, and cryo-electron microscopy with modelling tools, the authors conclude that MFAP4 assembles as octamers in the presence of calcium ions, but the octamers dissociate into tetramers in calcium-depleted conditions. A mutant MFAP4(C34S) was also studied to analyze the effect of disulfide bonds on the protein structure and stability. The binding of MFAP4 and MFAP4(C34S) to elastogenic proteins and cells (human dermal fibroblasts) were studied to understand how the structure of MFAP4 and the conditions (presence or absence of calcium ions) affect the interactions with elastogenic proteins and cells. The manuscript presents a rigorous experimental work and a detailed structural analysis that are relevant to better understand the function of MFP4 in elastogenesis, and it is also an important base for further research on the role of MFP4 in different diseases. The work is of interest for the readers of Nature Communications, but a minor revision is needed:

- 1) Supplementary Figures S2 and S3 are not referred to or commented in the main manuscript.

Supplemental Figure S2 is now referred to on line 144.
Supplemental Figure S3 is now referred to on line 191.

- 2) Figures 3B and 3C: it should be explained in the figure caption that each curve corresponds to a different protein concentration.

We amended the caption for Fig. 3 as requested and included the protein concentrations (lines 845-846) and also for Fig. 4 as requested by reviewer 2 (lines 863-865). We also included in caption of Fig. 3 several aspects that were requested by reviewer 2 (please see comment 8) (lines 846-849).

- 3) Figure 3G presents results from gel filtration chromatography, where 3 peaks are observed. The authors should elaborate the description in lines 250-254 a bit more and explain what oligomers are associated to each peak.

This comment is in part similar to comment 5 of reviewer 2. Based on the reviewer's comment, we performed additional analyses of gel filtration fractions by DLS and AFM. Surprisingly, we found that the first elution peak around ~6.4-6.5 mL did not contain Ca²⁺-loaded octamers as expected, rather than very large globular aggregates in the range of 35 nm hydrodynamic radii corresponding to ~1,000-10,000 kDa (**Reviewer Fig. 1**; shown below for comment 5 of reviewer 2). Since such large globular aggregates were not present in any other analysis throughout the entire manuscript (DLS, AFM, TEM, cryo-EM), we conclude that those must be an artefact of the gel filtration experiment. On the other hand, Ca²⁺-loaded octamers were enriched in the peak at ~8.5-8.6 mL, and Ca²⁺-depleted tetramers in the peak at ~9.2-9.5 mL. Since we cannot explain why the gel filtration produces these large globular aggregates that are not detectable by any other method, we decided to remove the gel filtration experiment from the paper. It did not provide essential information for the conclusions of the paper.

4) Some of the SPR sensograms in Figure 4 show a decrease with time, suggesting detachment of bound proteins (for instance, tropoelastin, fibulin-4 and fibulin-5 interacting with MFAP4 in the presence of calcium ions). Could the authors comment on that?

The SPR assays were executed by injecting the analyte over the sensor-bound ligand for 3 min, allowing binding of the analyte to the ligand (association period characterized by the k_a rate). This is followed by monitoring the dissociation of the analyte from the ligand for 10 min using only buffer without the analyte (dissociation period characterized by the k_d rate). Therefore, a decrease of the SPR signal over time represents the dissociation of the analyte bound to the ligand. This is true for every interaction and is part of the procedure to determine the dissociation constants (K_D) as a measurement for affinity. The lower the K_D , the higher the affinity of the ligands to each other, and vice versa. For tropoelastin, fibulin-4, and fibulin-5, the k_a rate is relatively high compared to other stronger interactions, resulting in the reported K_D values of 45 nM, 110 nM, and 357 nM, respectively. We have included more details of the SPR method on lines 623-646. We have also included the curve fittings onto the sensorgrams and the χ^2 values as a measurement for the goodness of these fits. Please also see additional details explained in our reply to comment 8 of reviewer 2.

5) In lines 412-415 the authors state: “The data suggest that the interactions of MFAP4 with fibulin-5 and with the LTBP4 isoforms depend on stabilisation of the pole regions by inter-molecular disulfide bonds, but other interactions do not require this stabilisation.” However, Figure 4 shows that MFAP4(C34S) can also bind LTBP4 isoforms in the presence of calcium ions, in contradiction with the previous statement.

This was indeed not described correctly, and we appreciate catching this error. We have now modified the text on line 426-427.

REVIEWER #2:

This manuscript describes the 3.55 Angstrom cryoEM structure of the octameric MFAP4, an extracellular protein linked to disease and pathological changes such as fibrosis. From their structure, the authors identify a number of new, interesting features; this includes a role for divalent cation binding in the formation of the MFAP4 octamer, coordinated by residues around Asp 191. They also show that cysteine residue 34 is important for the formation of an intermolecular disulfide bond but which is not required for multimerization of MFAP4. They also show that some protein interactions require divalent cations. The cryoEM data and resulting structure appears to be of high quality, but I had some queries and comments about the manuscript that I hope the authors can address.

1) The cryoEM map enabled the authors to resolve the positions of residues Q37-A255 but the amino acid residues preceding Q37 were not observed in the cryoEM map. Despite this, the loop at the N-terminus is modelled.

Is it appropriate to model this loop and the disulfide bond if the density isn't observed for this region in the map? If the disulfide bond at C34 has formed why are residues 34-37 disordered and not constrained?

We agree with the Reviewer's comment. Consequently, we have removed residues 34-37 from our model as they are not sufficiently constrained by our cryo-EM data to technically support this modelling.

We have maintained our representation of C34 disulfide bonds in Fig. 2, S3 and S4 as an integrative model with notes in the figure legends to alert the reader to the fact that residues 34-37 have been interpolated from a straightforward combination of cryo-EM data and supporting biochemical evidence. We feel that modelling intermolecular C34 disulfide bonds is appropriate in this context, given the clear evidence of

the SDS-PAGE that these residues are disulfide-bonded. It is highly likely that the flexibility of L35 and Q36 prevents the resolution of disulfide-bonded C34 residues within our post-processed cryo-EM map, and we provide this as a plausible explanation for “not-seeing” densities from these residues in the map.

We have amended the main text with new edits/modifications in lines 149-151 and 160-163. We have amended the legend for Fig. 2 (lines 827-833). We have also modified the legend for Movie 1.

2) Is the cryoEM data deposited in EMDB/EMPIAR, I didn't spot any accession numbers?

The data has been deposited to Electron Microscopy Data Bank (www.ebi.ac.uk/emdb; accession numbers EMD-42394 and EMD-42398 and the Protein Data Bank (<https://www.rcsb.org>; accession number 8UN7). The data will not be uploaded to EMPIAR. These accession numbers have been included in the methods section under “Cryo-electron microscopy, single-particle analysis and atomic modelling” (lines 582, 601, 610).

3) It is indicated that “relatively low levels of DTT (~3 mM) were required to reduce MFAP4 to monomers which supported a surface-located position for the intermolecular C34 disulfide bond”. 3mM DTT seem a high concentration when compared to the likely protein concentration. How does 3mM compare to the concentration of MFAP4 in this experiment and would this concentration of DTT be expected to reduce inter vs intramolecular disulfide bonds?

We apologize for not making this procedure sufficiently clear in the original manuscript. This experiment was principally performed as follows:

1. MFAP4 was incubated with decreasing DTT concentrations under physiological buffer conditions (TBS/2mM Ca²⁺).
2. The surplus DTT and reduced cysteine residues were non-reversibly modified by adding an excess of iodoacetamide (still under physiological buffer conditions).
3. The protein was run on SDS gel without further addition of DTT.

Generally, it is possible to limit the reduction of disulfide-bonds to those in exposed positions in the native protein. Disulfide-bonds holding together two or more polypeptide chains are often accessible to solvents and reagents without unfolding the polypeptide chain, whereas intrachain disulfide bridges are usually inaccessible. This makes it possible to selectively reduce the exposed disulfide bonds without affecting those in the interior. A typical procedure is described for reducing the interchain disulfide bridges in an immunoglobulin (Konigsberg W. (1972) Reduction of disulfide bonds in proteins with dithiothreitol. *Methods Enzymol* 25, 185-188)). The important aspect of the procedure used in our approach was exposing MFAP4 to DTT in a physiological buffer, not in a denaturing buffer (e.g. SDS, guanidine HCl) that would expose the intrachain disulfide-bonds. This aspect and the moderate concentration of 3 mM DTT (which is still ~40-fold in excess over the disulfide-bonds in MFAP4) make us confident that we reduce intermolecular disulfide bonds with this approach.

We have modified lines 121 and 164-165 in the main text, lines 512-517 in the methods, and the legend for Supplemental Figure 1C to clarify these aspects.

4) The link to calcium is made as other FReD proteins bind calcium, and EDTA treatment of MFAP4 abolishes some interactions and decreases the size of the octamer. Why was EDTA used to chelate metal ions rather than the more calcium specific EGTA? Can the authors rule out the contribution of a different divalent cation, or show that calcium is indeed the bound metal ion in the cryoEM density? Perhaps titrating back in calcium specifically to monitor octamer formation and binding. A more elegant experiment might be to mutate the Aspartate residues that appear to coordinate the metal ion to see if this abolishes both octamer formation and metal-ion binding.

Both EDTA and EGTA can effectively remove Ca^{2+} ions from proteins. However - as the reviewer pointed out - EGTA is more specific for Ca^{2+} than EDTA. In response to the reviewer's comments, we have conducted a series of additional experiments with MFAP4 and MFAP4_{C34S} using EGTA and included some of them in the revised figures. We have also included titration experiments with other divalent cations. In Fig. 3A, we now show the DLS results of MFAP4 and MFAP4_{C34S} treated with EDTA or EGTA. The results are virtually identical. In Fig. 3B, we included self-interaction of MFAP4 and MFAP4_{C34S} after EGTA treatment and replenishment with Mg^{2+} , showing the absence of self-interaction. In the new Supplemental Fig. S7, we tested the interactions of MFAP4 and MFAP4_{C34S} with two selected elastogenic proteins (rFBN1-N, tropoelastin) in the presence of Mg^{2+} . As expected, the Ca^{2+} -dependent interactions with rFBN1-N and tropoelastin are absent in the presence of Mg^{2+} . In Figure 3D, we titrated either Ca^{2+} or Mg^{2+} to EGTA-treated MFAP4 and MFAP4_{C34S} and determined the hydrodynamic radii by DLS. Only Ca^{2+} titration resulted in octamer formation from tetramers, not Mg^{2+} titration. In the new Supplemental Fig. S5, we added other divalent cations (Zn^{2+} , Cu^{2+} , Mn^{2+}) at their respective physiological concentrations after EGTA-treatment of MFAP4 and MFAP4_{C34S}. None of these cations triggered octamer formation; the proteins remained tetrameric. Together, these data clearly demonstrate that the octamer formation of MFAP4 and MFAP4_{C34S} depends on Ca^{2+} and not on other divalent cations.

In addition to the figure changes explained above, we modified the figure legends accordingly and amended the main text and the methods on lines 227-228, 232-233, 237-245, 286-289, and 530-536.

5) Following this point, metal ion binding appears to be a requirement for interaction with some proteins shown with SPR. It is not clear from the methods whether gel filtration was performed on samples prior to SPR and if the preparation of MFAP4 for both the WT and cys mutant is a mixture of oligomeric states (as suggested in the gel filtration analysis in Fig 3G). I don't think it is possible then to say that only the octameric form binds given the data presented, metal-ion binding is certainly required for the interaction and metal-ion binding stabilises the oligomers but decoupling these factors requires more subtlety.

On a related point, generally from the methods it is very difficult to determine how the protein was purified prior to cryoEM, SPR or DLS. Was gel filtration performed as a final purification step after affinity chromatography and what was the oligomeric state of MFAP4 in these experiments, ie purified octamer or a heterogeneous mixture as shown in Fig 3G prior to gel filtration in the presence of metal ions? For example, the DLS methods state "Samples of different MFAP4 and MFAP4C34S preparations in various buffer conditions, either gel filtrated or not, or treated with EDTA were analysed in quadruplicates and the results were averaged." The details of whether gel filtered protein or what buffer used is not in the figure legends, so this is not clear.

Based on the reviewer's comment, we performed additional analyses of gel filtration fractions by DLS and AFM. Surprisingly, we found that the first elution peak around ~6.4-6.5 mL did not contain Ca^{2+} -loaded octamers as expected, rather than very large globular aggregates in the range of 35 nm hydrodynamic radii corresponding to ~1,000-10,000 kDa (Reviewer Fig. 1). Since such large globular aggregates were not present in any other analysis throughout the entire manuscript (DLS, AFM, TEM, cryo-EM), we conclude that those must be an artifact of the gel filtration experiment. On the other hand, Ca^{2+} -loaded octamers were enriched in the peak at ~8.5-8.6 mL, and Ca^{2+} -depleted tetramers in the peak at ~9.2-9.5 mL. Since we cannot explain why the gel filtration produces these large globular aggregates that are not detectable by any other method, we decided to remove the gel filtration experiment from the paper. It did not provide essential information for the conclusions of the paper.

Reviewer Fig. 1 Gel filtration of MFAP4 and MFAP4_{C34S} in TBS either in the presence (+Ca²⁺) or absence (-Ca²⁺; EDTA treated) of calcium (top left graph). DLS analysis (bottom left) of selected fractions to determine the MFAP4 particle sizes as indicated by the coloured bars. AFM images (right panel) for visualization of the MFAP4 particles of the selected fractions measured in DLS.

All SPR, DLS, AFM, TEM and cryo-EM experiments were performed with non-gel-filtrated proteins. The combined results from DLS, AFM, TEM, and cryo-EM all support that the dominant particles for Ca²⁺-loaded MFAP4 and MFAP4_{C34S} are octamers. For the Ca²⁺-depleted proteins (either by EDTA or EGTA) or the proteins in the presence of divalent metal ions other than Ca²⁺, all data support an enrichment of tetrameric particles.

We describe now in more detail in the Methods how the proteins were specifically prepared for SPR experiments (lines 623-646), DLS (lines 530-536), and cryo-EM (lines 565-567).

6) Interestingly in Fig 1F, increasing NaCl concentration causes the hydrodynamic radius of MFAP4_{C34S} to reduce to a similar size as the calcium absent form, which is thought to be a tetramer (Fig3A). Why does changing salt concentration not cause the wildtype MFAP4 to dissociate to a tetramer if disulfide bonding does not stabilise the octameric form?

These data indicate a role of the C34-mediated intermolecular disulfide bond in stabilising higher order assembly, without being strictly required, as the hydrodynamic radius of the assembled complex decreases more for MFAP_{C34S} than for MFAP4 with increasing NaCl concentrations.

Part of this statement was previously included in the discussion on lines 358-361. We now amended this statement on line 359.

7) For figure 3, the TEM of the non-calcium bound structure does not show definitively that it is a tetramer due to the defined orientation and lack of side views. It would be nice to see some 2D classification of the side views, not just the top views, to show it is a tetramer. Were side-views present in negative stained images, and were attempts made to try different cryoEM grids, add a carbon support film or collect tilted data to optimise the conditions?

We thank the reviewer for these comments. We would like to clarify that side views were not present in 2D classes, as indicated in Supplemental Fig. 6. We collected a second cryoEM dataset of MFAP4 without Ca²⁺ with the focus to acquire thicker regions of vitrified material with the rationale that thicker regions would allow MFAP4 tetramers to adopt more orientations. However, even with this larger dataset

including thicker regions of vitrified material, our 3D reconstructions suffered from preferential orientation. Although it may be possible to optimise grid conditions using different supports or by collecting tilted data for further analysis, we do not think this would substantially change our interpretation that MFAP4 assembles as a tetramer in the absence of Ca^{2+} .

Trials of negative staining TEM also failed to produce side-views of the proteins, likely due to the preferential orientation of the molecules when absorbed onto the continuous carbon surface of the TEM grid. Indeed, the fact that the molecules adopt this conformation favours that it is a tetramer. We can image it as a rectangular prism-like object with two large faces and four more narrow faces that impede positioning/anchoring on the horizontal surface, possibly falling flat with the larger faces on the continuous carbon surface.

This notion is further supported by AFM analyses that allows us to determine the height of the particles. While all particles in AFM analyses appear similar than in TEM after negative staining, the particles are higher in the Ca^{2+} -saturated versus the Ca^{2+} -free form. This is consistent with the idea that tetramers never land on the side rather than on the top/bottom, but the height of the tetramers is significantly lower than that of octamers. We have included these AFM data in the new Fig. 3G and amended the main text on lines 258-262.

8) For figure 4, all SPR traces should include a key with the different concentration of analyte used (information also not in legend in fig 3). The methods indicate that a 1:1 Langmuir model was used for kinetic fitting, and this fit should be overlaid onto all kinetic traces along with chi-squared values for these fits (even if this is included in the supplementary material). Certainly, for some SPR traces they do not visually appear to represent a 1:1 binding fit and there are some unusual features in these data, for example MFAP4C34S with Ca^{2+} binding to tropoelastin it is hard to distinguish the association/dissociation phases.

All SPR interactions were performed with the following analyte concentrations: 100, 50, 20, 10, 5, and 0 $\mu\text{g}/\text{mL}$. We are stating this now in the Methods on lines 630-632, in the legend of Fig. 3 on lines 845-846, and Fig. 4 on lines 862-864.

The fittings were performed with the instrument-integrated module “*Fit Kinetics Separate*” (BIAevaluation software). This software tool performs individual analysis of the association (k_a) and dissociation (k_d) profiles. The K_D dissociation constants were calculated as the k_d/k_a ratio and averaged over all analyte concentrations. The mean K_D and its standard deviation are reported on each sensorgram.

MFAP4/MFAP4_{C34S} were immobilized on the chips. We do not know whether one or more subunits participate in the interaction with the soluble elastogenic proteins. We also do not know whether the respective elastogenic proteins have one or more binding sites for MFAP4, or whether glycosylation sites may affect the interaction. Without knowing all these details, we employed the most widely used 1:1 Langmuir binding model. This model is frequently used in the context of extracellular matrix interactions; in most published cases, the specific stoichiometry is unknown. However, the Biacore BIAevaluation Software Handbook (version 4, edition June 2004, page A-6) describes χ^2 values < 10 as acceptable to consider the fitting sufficiently accurate. In our analyses, most χ^2 values were < 2 and many even < 1 , indicating excellent fits. For a few analyses, the χ^2 values were between 2-5, but, again, still acceptable as a reasonable fit. Given that association and dissociation curves were fitted separately, we would have to report 160 χ^2 values for Fig. 4 alone. Since this is not very practical, we have decided to include for each graph a χ^2 value range indicating the smallest and the largest χ^2 value. We have now overlaid the curve fittings in yellow on all kinetic traces in Figures 3 and 4. For a few dissociation traces that showed a drift over time to higher RU values, the fitting was executed using only the descending portion of the respective trace. Associations with negligible signal changes were omitted for fitting.

Regarding Ca²⁺-loaded MFAP4_{C34S} binding to tropoelastin, we have now included a mark that clearly distinguishes the association/dissociation phases.

We updated the Methods under “Protein and cell binding assays” (lines 623-646) and the legends of Figures 3 (lines 834-836) and 4 (lines 652-654) to better describe the explained aspects.

9) For figure 5, there is very little detail about the data processing of the higher order oligomeric assemblies. The authors should include 3D maps of the oligomeric reconstructions and at least 2D classes to show the classification of the larger repeats.

We thank the reviewer for drawing our attention to the need for clarification for Fig. 6 (not Fig. 5 as stated). We have now included the 3D map of the oligomeric reconstruction. We have now used 2D classification on the complete dataset that contributed to the high-resolution map of MFAP4 with Ca²⁺ (444,005 particles) to show that 173,879 particles have a nearby octamer. We have modified significantly Fig. 6 and its legend. Further amendments were included in the main text on lines 322-325 and 334-335, and in the Methods on lines 584-603.

Minor points:

10) State the box size or scale for the 2D classes in Figure 2.

Figure 2 legend has been updated that the 2D classes were produced from particles extracted with a box size of 273.6 Å. Further amendments were made in the Methods on lines 576-577 and 581-600.

11) In Figure 3, the panels should flow in alphabetical order, currently it is ABCGFDE.

We have modified Figure 3 and added new results and subfigures, and the letters are now in alphabetical order.

Additional changes not requested by the reviewers:

We have included several minor improvements to the manuscript to further clarify some aspects. These amendments did not change any interpretation or conclusion. They are on lines 235, 255, 265, 267, 283-284, 285, 294, 512-517, 544-548, 557-559, 607, 610, 652, and 653-654.

REVIEWERS' COMMENTS

Reviewer #1 (Remarks to the Author):

The authors have addressed all the comments properly. The revised manuscript can be published in Nature Communications.

Reviewer #2 (Remarks to the Author):

All my queries have been satisfactorily addressed by the authors.